# GRADIENT STEP DENOISER FOR CONVERGENT PLUG-AND-PLAY

**Samuel Hurault** *, **Arthur Leclaire & Nicolas Papadakis**
Univ. Bordeaux, Bordeaux INP, CNRS, IMB, UMR 5251,F-33400 Talence, France

## ABSTRACT

Plug-and-Play (PnP) methods constitute a class of iterative algorithms for imaging problems where regularization is performed by an off-the-shelf denoiser. Although PnP methods can lead to tremendous visual performance for various image problems, the few existing convergence guarantees are based on unrealistic (or suboptimal) hypotheses on the denoiser, or limited to strongly convex data-fidelity terms. We propose a new type of PnP method, based on half-quadratic splitting, for which the denoiser is realized as a gradient descent step on a functional parameterized by a deep neural network. Exploiting convergence results for proximal gradient descent algorithms in the nonconvex setting, we show that the proposed PnP algorithm is a convergent iterative scheme that targets stationary points of an explicit global functional. Besides, experiments show that it is possible to learn such a deep denoiser while not compromising the performance in comparison to other state-of-the-art deep denoisers used in PnP schemes. We apply our proximal gradient algorithm to various ill-posed inverse problems, e.g. deblurring, super-resolution and inpainting. For all these applications, numerical results empirically confirm the convergence results. Experiments also show that this new algorithm reaches state-of-the-art performance, both quantitatively and qualitatively.

## 1 INTRODUCTION

Image restoration (IR) problems can be formulated as inverse problems of the form

$$x^* \in \arg\min_x f(x) + \lambda g(x) \tag{1}$$

where $f$ is a term measuring the fidelity to a degraded observation $y$, and $g$ is a regularization term weighted by a parameter $\lambda \geq 0$. Generally, the degradation of a clean image $\hat{x}$ can be modeled by a linear operation $y = A\hat{x} + \xi$, where $A$ is a degradation matrix and $\xi$ a white Gaussian noise. In this context, the maximum a posteriori (MAP) derivation relates the data-fidelity term to the likelihood $f(x) = -\log p(y|x) = \frac{1}{2\sigma^2}||Ax - y||^2$, while the regularization term is related to the chosen prior.

Regularization is crucial since it tackles the ill-posedness of the IR task by bringing a priori knowledge on the solution. A lot of research has been dedicated to designing accurate priors $g$. Among the most classical priors, one can single out total variation (Rudin et al., 1992), wavelet sparsity (Mallat, 2009) or patch-based Gaussian mixtures (Zoran & Weiss, 2011). Designing a relevant prior $g$ is a difficult task and recent approaches rather apply deep learning techniques to directly learn a prior from a database of clean images (Lunz et al., 2018; Prost et al., 2021; González et al., 2021).

Generally, the problem (1) does not have a closed-form solution, and an optimization algorithm is required. First-order proximal splitting algorithms (Combettes & Pesquet, 2011) operate individually on $f$ and $g$ via the proximity operator

$$\text{Prox}_f(x) = \arg\min_z \frac{1}{2}||x - z||^2 + f(z). \tag{2}$$

Among them, half-quadratic splitting (HQS) (Geman & Yang, 1995) alternately applies the proximal operators of $f$ and $g$. Proximal methods are particularly useful when either $f$ or $g$ is nonsmooth.

Plug-and-Play (PnP) methods (Venkatakrishnan et al., 2013) build on proximal splitting algorithms by replacing the proximity operator of $g$ with a generic denoiser, *e.g.* a pretrained deep network.

---

*Corresponding author: samuel.hurault@math.u-bordeaux.fr

These methods achieve state-of-the-art results (Buzzard et al., 2018; Ahmad et al., 2020; Yuan et al., 2020; Zhang et al., 2021) in various IR problems. However, since a generic denoiser cannot generally be expressed as a proximal mapping (Moreau, 1965), convergence results, which stem from the properties of the proximal operator, are difficult to obtain. Moreover, the regularizer $g$ is only made implicit via the denoising operation. Therefore, PnP algorithms do not seek the minimization of an explicit objective functional which strongly limits their interpretation and numerical control.

In order to keep tractability of a minimization problem, Romano et al. (2017) proposed, with regularization by denoising (RED), an explicit prior $g$ that exploits a given generic denoiser $D$ in the form $g(x) = \frac{1}{2}\langle x, x - D(x) \rangle$. With strong assumptions on the denoiser (in particular a symmetric Jacobian assumption), they show that it verifies

$$\nabla_x g(x) = x - D(x). \tag{3}$$

Such a denoiser is then plugged in gradient-based minimization schemes. Despite having shown very good results on various image restoration tasks, as later pointed out by Reehorst & Schniter (2018) or Saremi (2019), existing deep denoisers lack Jacobian symmetry. Hence, RED does not minimize an explicit functional and is not guaranteed to converge.

**Contributions.** In this work, we develop a PnP scheme with novel theoretical convergence guarantees and state-of-the-art IR performance. Departing from the PnP-HQS framework, we plug a denoiser that inherently satisfies equation (3) without sacrificing the denoising performance. The resulting fixed-point algorithm is guaranteed to converge to a stationary point of an explicit functional. This convergence guarantee does not require strong convexity of the data-fidelity term, thus encompassing ill-posed IR tasks like deblurring, super-resolution or inpainting.

## 2 RELATED WORKS

PnP methods have been successfully applied in the literature with various splitting schemes: HQS (Zhang et al., 2017b; 2021), ADMM (Romano et al., 2017; Ryu et al., 2019), Proximal Gradient Descent (PGD) (Terris et al., 2020). First used with classical non deep denoisers such as BM3D (Chan et al., 2016) and pseudo-linear denoisers (Nair et al., 2021; Gavaskar et al., 2021), more recent PnP approaches (Meinhardt et al., 2017; Ryu et al., 2019) rely on efficient off-the-shelf deep denoisers such as DnCNN (Zhang et al., 2017a). State-of-the-art IR results are currently obtained with denoisers that are specifically designed to be integrated in PnP schemes, like IR-CNN (Zhang et al., 2017b) or DRUNET (Zhang et al., 2021). Though providing excellent restorations, such schemes are not guaranteed to converge for all kinds of denoisers or IR tasks.

Designing convergence proofs for PnP algorithms is an active research topic. Sreehari et al. (2016) used the proximal theorem of Moreau (Moreau, 1965) to give sufficient conditions for the denoiser to be an explicit proximal map, which are applied to a pseudo-linear denoiser. The convergence with pseudo-linear denoisers have been extensively studied (Gavaskar & Chaudhury, 2020; Nair et al., 2021; Chan, 2019). However, state-of-the-art PnP results are obtained with deep denoisers. Various assumptions have been made to ensure the convergence of the related PnP schemes. With a "bounded denoiser" assumption, Chan et al. (2016); Gavaskar & Chaudhury (2019) showed convergence of PnP-ADMM with stepsizes decreasing to 0. RED (Romano et al., 2017) and RED-PRO (Cohen et al., 2021) respectively consider the classes of denoisers with symmetric Jacobian or demicontractive mappings, but these conditions are either too restrictive or hard to verify in practice. In Appendix A.3, more details are given on RED-based methods. Many works focus on Lipschitz properties of PnP operators. Depending on the splitting algorithm in use, convergence can be obtained by assuming the denoiser averaged (Sun et al., 2019b), firmly nonexpansive (Sun et al., 2021; Terris et al., 2020) or simply nonexpansive (Reehorst & Schniter, 2018; Liu et al., 2021). These settings are unrealistic as deep denoisers do not generally satisfy such properties. Ryu et al. (2019); Terris et al. (2020) propose different ways to train deep denoisers with constrained Lipschitz constants, in order to fit the technical properties required for convergence. But imposing hard Lipschitz constraints on the network alters its denoising performance (Bohra et al., 2021; Hertrich et al., 2021). Yet, Ryu et al. (2019) manages to get a convergent PnP scheme without assuming the nonexpansiveness of $D$. This comes at the cost of imposing strong convexity on the data-fidelity term $f$, which excludes many IR tasks like deblurring, super-resolution or inpainting. Hence, given the ill-posedness of IR problems, looking for a unique solution via contractive operators is a restrictive assumption. In this work, we do not impose contractiveness, but still obtain convergence results with realistic hypotheses.

One can relate the ideal deep denoiser to the "true" natural image prior $p$ via Tweedie's Identity. In (Efron, 2011), it is indeed shown that the Minimum Mean Square Error (MMSE) denoiser $D_\sigma^*$ (at noise level $\sigma$) verifies $D_\sigma(x) = x + \sigma^2 \nabla_x \log p_\sigma(x)$ where $p_\sigma$ is the convolution of $p$ with the density of $\mathcal{N}(0, \sigma^2 \, \mathrm{Id})$. In a recent line of research (Bigdeli et al., 2017; Xu et al., 2020; Laumont et al., 2021; Kadkhodaie & Simoncelli, 2020), this relation is used to plug a denoiser in gradient-based dynamics. In practice, the MMSE denoiser cannot be computed explicitly and Tweedie's Identity does not hold for deep approximations of the MMSE. In order to be as exhaustive as possible, we detailed the addressed limitations of existing PnP methods in Appendix A.1.

## 3 THE GRADIENT STEP PLUG-AND-PLAY

The proposed method is based on the PnP version of half-quadratic-splitting (PnP-HQS) that amounts to replacing the proximity operator of the prior $g$ with an off-the-shelf denoiser $D_\sigma$. In order to define a convergent PnP scheme, we first set up in Section 3.1 a Gradient Step (GS) denoiser. We then introduce the Gradient Step PnP (GS-PnP) algorithm in Section 3.2.

### 3.1 GRADIENT STEP DENOISER

We propose to plug a denoising operator $D_\sigma$ that takes the form of a gradient descent step

$$D_\sigma = \mathrm{Id} - \nabla g_\sigma, \tag{4}$$

with $g_\sigma : \mathbb{R}^n \to \mathbb{R}$. Contrary to Romano et al. (2017), our denoiser exactly represents a conservative vector field. The choice of the parameterization of $g_\sigma$ is fundamental for the denoising performance. As already noticed in Salimans & Ho (2021), we experimentally found that directly modeling $g_\sigma$ as a neural network (*e.g.* a standard network used for classification) leads to poor denoising performance. In order to keep the strength of state-of-the-art unconstrained denoisers, we rather use

$$g_\sigma(x) = \frac{1}{2}||x - N_\sigma(x)||^2, \tag{5}$$

$$\text{which leads to } D_\sigma(x) = x - \nabla g_\sigma(x) = N_\sigma(x) + J_{N_\sigma}(x)^T(x - N_\sigma(x)), \tag{6}$$

where $N_\sigma : \mathbb{R}^n \to \mathbb{R}^n$ is parameterized by a neural network and $J_{N_\sigma}(x)$ is the Jacobian of $N_\sigma$ at point $x$. As discussed in Appendix A.2, the formulation (5) for $g_\sigma$ has been proposed in (Romano et al., 2017, Section 5.2) and (Bigdeli & Zwicker, 2017) for a distinct but related purpose, and not exploited for convergence analysis. Thanks to our definition (6) for $D_\sigma$, **we can parameterize $N_\sigma$ with any differentiable neural network architecture** $\mathbb{R}^n \to \mathbb{R}^n$ that has proven efficient for image denoising. Although the representation power of the denoiser is limited by the particular form (6), we show (see Section 5.1) that such parameterization still yields state-of-the-art denoising results. We train the denoiser $D_\sigma$ for Gaussian noise by minimizing the MSE loss function

$$\mathcal{L}(D_\sigma) = \mathbb{E}_{x \sim p, \xi_\sigma \sim \mathcal{N}(0, \sigma^2 I)}[||D_\sigma(x + \xi_\sigma) - x||^2], \tag{7}$$

$$\text{or } \mathcal{L}(g_\sigma) = \mathbb{E}_{x \sim p, \xi_\sigma \sim \mathcal{N}(0, \sigma^2 I)}[||\nabla g_\sigma(x + \xi_\sigma) - \xi_\sigma||^2], \tag{8}$$

when written in terms of $g_\sigma$ using equation (4).

**Remark 1.** *By definition, the optimal solution $g_\sigma^* \in \arg\min \mathcal{L}$ is related to the MMSE denoiser $D_\sigma^*$, that is, the best non-linear predictor of $x$ given $x + \xi_\sigma$. Therefore, it satisfies Tweedie's formula and $\nabla g_\sigma^* = -\sigma^2 \nabla \log p_\sigma$ (Efron, 2011) i.e. $g_\sigma^* = -\sigma^2 \log p_\sigma + C$, for some $C \in \mathbb{R}$. Hence approximating the MMSE denoiser with a denoiser parameterized as (4) is related to approximating the logarithm of the smoothed image prior of $p_\sigma$ with $-\frac{1}{\sigma^2}g_\sigma$. This relation was used for image generation with "Denoising Score Matching" by Saremi & Hyvarinen (2019); Bigdeli et al. (2020).*

### 3.2 A PLUG-AND-PLAY METHOD FOR EXPLICIT MINIMIZATION

The standard PnP-HQS operator is $T_{\text{PnP-HQS}} = D_\sigma \circ \mathrm{Prox}_{\tau f}$, *i.e.* $(\mathrm{Id} - \nabla g_\sigma) \circ \mathrm{Prox}_{\tau f}$ when using the GS denoiser as $D_\sigma$. For convergence analysis, we wish to fit the proximal gradient descent (PGD) algorithm. We thus propose to switch the proximal and gradient steps and to relax the denoising step with a parameter $\lambda \geq 0$. Our PnP algorithm with GS denoiser (GS-PnP) then writes

$$x_{k+1} = T_{\text{GS-PnP}}^{\tau, \lambda}(x_k) \text{ with } \begin{aligned} T_{\text{GS-PnP}}^{\tau, \lambda} &= \mathrm{Prox}_{\tau f} \circ (\tau \lambda D_\sigma + (1 - \tau \lambda) \, \mathrm{Id}), \\ &= \mathrm{Prox}_{\tau f} \circ (\mathrm{Id} - \tau \lambda \nabla g_\sigma). \end{aligned} \tag{9}$$

Under suitable conditions on $f$ and $g_\sigma$ (see Lemma 1 in Appendix C), fixed points of the PGD operator $T_{\text{GS-PnP}}^{\tau,\lambda}$ correspond to critical points of a classical objective function in IR problems

$$F(x) = f(x) + \lambda g_\sigma(x). \tag{10}$$

Therefore, using the GS denoiser from equation (4) is equivalent to include an explicit regularization and thus leads to a tractable global optimization problem solved by the PnP algorithm. Our complete PnP scheme is presented in Algorithm 1. It includes a backtracking procedure on the stepsize $\tau$ that will be detailed in Section 4.2. Also, after convergence, we found it useful to apply an extra gradient step $\text{Id} - \lambda\tau\nabla g_\sigma$ in order to discard the residual noise brought by the last proximal step $\text{Prox}_{\tau f}$.

## 4 CONVERGENCE ANALYSIS

In this section, we introduce conditions on $f$ and $D_\sigma$ that will ensure the convergence of the PnP iterations (9) towards a solution of (10). For that purpose, we make use of the literature of convergence analysis (Attouch et al., 2013; Beck, 2017; Beck & Teboulle, 2009) of the PGD algorithm in the nonconvex setting.

### 4.1 CONVERGENCE RESULTS

A common setting for image restoration is the convex smooth $L^2$ data-fidelity $f(x) = ||Ax - y||^2$. In order to cover the

---

**Algorithm 1:** Plug-and-Play image restoration
**Param.:** init. $z_0$, $\lambda > 0$, $\sigma \geq 0$, $\epsilon > 0$, $\tau_0 > 0$,
    $K \in \mathbb{N}^*$, $\eta \in (0,1)$, $\gamma \in (0, 1/2)$.
**Input** : degraded image $y$.
**Output:** restored image $\hat{x}$.
$k = 0$; $x_0 = \text{Prox}_{\tau f}(z_0)$; $\tau = \tau_0/\eta$; $\Delta > \epsilon$ ;
**while** $k < K$ and $\Delta > \epsilon$ **do**
    $z_k = \lambda\tau D_\sigma(x_k) + (1 - \lambda\tau)x_k$;
    $x_{k+1} = \text{Prox}_{\tau f}(z_k)$;
    **if** $F(x_k) - F(x_{k+1}) < \frac{\gamma}{\tau}||x_k - x_{k+1}||^2$ ;
    **then** $\tau = \eta\tau$;
    **else** $\Delta = \frac{F(x_k) - F(x_{k+1})}{F(x_0)}$; $k = k + 1$ ;
**end**
$\hat{x} = \lambda\tau D_\sigma(x_K) + (1 - \lambda\tau)x_K$;

---

noiseless case or to deal with a broader range of common degradation models, like Laplace or Poisson noise, we only assume $f$ to be proper, lower semicontinous and convex. Next, the regularizer $g_\sigma$ is assumed to be differentiable with Lipschitz gradient, but not necessarily convex. This assumption on $g_\sigma$ is reasonable from a practical perspective. Indeed, using a network $N_\sigma$ with differentiable activation functions, the function $g_\sigma$ introduced in Section 3 is differentiable with Lipschitz gradient (details and proof are given in Appendix B). Without further assumptions on $f$ and $g_\sigma$, the following theorem establishes the convergence of both the objective function values and the residual for a large variety of IR tasks.

**Theorem 1** (Proof in Appendix C). *Let $f : \mathbb{R}^n \to \mathbb{R} \cup \{+\infty\}$ and $g_\sigma : \mathbb{R}^n \to \mathbb{R}$ be proper lower semicontinous functions with $f$ convex and $g_\sigma$ differentiable with $L$-Lipschitz gradient. Let $\lambda > 0$, $F = f + \lambda g_\sigma$ and assume that $F$ is bounded from below. Then, for $\tau < \frac{1}{\lambda L}$, the iterates $x_k$ given by the iterative scheme (9) verify*

*(i) $(F(x_k))$ is non-increasing and converges.*

*(ii) The residual $||x_{k+1} - x_k||$ converges to 0.*

*(iii) All cluster points of the sequence $(x_k)$ are stationary points of (10).*

**Remark 2.** *In the nonconvex setting, the quantity $\gamma_k = \min_{0 \leq i \leq k} ||x_{i+1} - x_i||^2$ is commonly used to analyze the convergence rate of the algorithm (Beck & Teboulle, 2009; Ochs et al., 2014). Following the proof of Theorem 1 in Appendix C, we can obtain $\gamma_k \leq \frac{1}{k} \frac{F(x_0) - \lim F(x_k)}{\frac{1}{2\tau} - \frac{L}{2}}$ that is to say a $\mathcal{O}(\frac{1}{k})$ convergence rate for the squared $L^2$ residual.*

**Remark 3.** *Even if most data-fidelity terms $f$ encountered in image restoration are convex, Theorem 1 can be extended to nonconvex $f$. The proof of (iii) requires technical adaptations that can be found in Li & Lin (2015, Theorem 1) (as the 1-Lipschitz property of $\text{Prox}_{\tau f}$ does not hold anymore). Such nonconvex $f$ appear for example in the context of phase retrieval (Metzler et al., 2018).*

We can further obtain convergence of the iterates by assuming that the generated sequence $(x_k)$ is bounded and that $f$ and $g_\sigma$ verify the Kurdyka-Lojasiewicz (KL) property. The boundedness of $(x_k)$ is discussed in Appendix D. The KL property (defined in Appendix E) has been widely used to study the convergence of optimization algorithms in the nonconvex setting (Attouch et al., 2010; 2013; Ochs et al., 2014). Very large classes of functions, in particular all the semi-algebraic functions, satisfy this property. In practice, in the extent of our analysis, the KL property is always satisfied.

**Theorem 2** (Proof in Attouch et al. (2013), Theorem 5.1). *Let $f : \mathbb{R}^n \to \mathbb{R} \cup \{+\infty\}$ and $g_\sigma : \mathbb{R}^n \to \mathbb{R}$ be proper lower semicontinuous functions with $f$ convex and $g_\sigma$ differentiable with $L$-Lipschitz gradient. Let $\lambda > 0$, $F = f + \lambda g_\sigma$ and assume that $F$ is bounded from below. Assume that $F$ verify the KL property. Suppose that $\tau < \frac{1}{\lambda L}$. If the sequence $(x_k)$ given by the iterative scheme (9) is bounded, then it converges, with finite length, to a critical point $x^*$ of $F$.*

**Remark 4.** *As explained in Attouch et al. (2013, Remark 5.2), the continuity of $f$ is not required since we use the "exact forward-backward splitting algorithm". We can thus deal with non-continuous data-fidelity terms, as it is the case with the inpainting application in Appendix J.3.*

A more detailed description of all the assumptions of Theorems 1 and 2 is given in Appendix F.

## 4.2 Backtracking to handle the Lipschitz constant of $\nabla g_\sigma$

The convergence of Algorithm 1 actually requires to control the Lipschitz constant of $\nabla g_\sigma$ only on a small subset of images related to $\{x_k\}$. Therefore, estimating $L$ for all images and setting the maximum stepsize $\tau$ accordingly will lead to sub-optimal convergence speed. In order to avoid small stepsizes, we use the backtracking strategy of Beck (2017, Chapter 10) and Ochs et al. (2014).

The convergence study in the proof of Theorem 1 is based on the sufficient decrease property of $F$ established in equation (33). Without knowing the exact Lipschitz constant $L$, backtracking aims at finding the maximal stepsize $\tau$ yielding the sufficient decrease property. Given $\gamma \in (0, 1/2)$, $\eta \in [0, 1)$ and an initial stepsize $\tau_0 > 0$, the following update rule on $\tau$ is applied at each iteration $k$:

$$\text{while} \quad F(x_k) - F(T_{\text{GS-PnP}}^{\tau,\lambda}(x_k)) < \frac{\gamma}{\tau}||T_{\text{GS-PnP}}^{\tau,\lambda}(x_k) - x_k||^2, \quad \tau \longleftarrow \eta\tau. \tag{11}$$

**Proposition 1** (Proof in Appendix G). *Under the assumptions of Theorem 1, at each iteration of the algorithm, the backtracking procedure (11) is finite (i.e. a stepsize satifying $F(x_k) - F(T_{GS\text{-}PnP}^{\tau,\lambda}(x_k)) \geq \frac{\gamma}{\tau}||T_{GS\text{-}PnP}^{\tau,\lambda}(x_k) - x_k||^2$ is found in a finite number if iterations), and with backtracking, the convergence results of Theorem 1 and Theorem 2 still hold.*

**Remark 5.** *In practice, as explained in Section 5, we choose to initialize the stepsize as $\lambda\tau_0 = 1$ and backtracking hardly ever activates. In this particular case, our results prove convergence of the standard PnP-HQS scheme $x_{k+1} = \text{Prox}_{\tau f} \circ D_\sigma(x_k)$ applied with our specific denoiser.*

## 5 Experiments

In this section, we first show the performance of the GS denoiser. Next we empirically confirm that our GS-PnP method is convergent while providing state-of-the art results for different IR tasks.

### 5.1 Gradient-descent-based denoiser

**Denoising Network Architecture** We choose to parameterize $N_\sigma$ with the architecture DRUNet (Zhang et al., 2021)) (represented in Appendix H), a U-Net in which residual blocks are integrated. One first benefit of DRUNet is that it is built to take the noise level $\sigma$ as input, which is consistent with our formulation. Also, the U-Net models have previously offered good results in the context of prior approximation via Denoising Score Matching (Ho et al., 2020). Furthermore, Zhang et al. (2021) showed that DRUNet yields state-of-the-art results for denoising but also for PnP image restoration. In order to ensure differentiability w.r.t the input, we change RELU activations to ELU. We also limit the number of residual blocks to 2 at each scale to lower the computational burden.

**Training details** We use the color image training dataset proposed in Zhang et al. (2021) *i.e.* a combination of the Berkeley segmentation dataset (CBSD) (Martin et al., 2001), Waterloo Exploration Database (Ma et al., 2017), DIV2K dataset (Agustsson & Timofte, 2017) and Flick2K dataset (Lim et al., 2017). During training, the input images are corrupted with a white Gaussian noise $\xi_\sigma$ with standard deviation $\sigma$ randomly chosen in $[0, 50/255]$. With our parameterization (6) of $D_\sigma$, the network $N_\sigma$ is trained to minimize the $L^2$ loss (7). While the original DRUNet is trained with $L^1$ loss, we stick to the $L^2$ loss to keep the interpretability of $g_\sigma$ as an approximation of the log prior (see Remark 1). For each batch, the gradient $\nabla g_\sigma$ is computed with PyTorch differentiation tools.

We train the model on $128 \times 128$ patches randomly sampled from the training images, with batch size 16, during 1500 epochs. We use the ADAM optimizer with learning rate $10^{-4}$, divided by 2 every 300 epochs. It takes around one week to train the model on a single Tesla P100 GPU.

**Denoising results**   We evaluate the PSNR performance of the proposed GS denoiser (GS-DRUNet) on $256 \times 256$ color images center-cropped from the original CBSD68 dataset images (Martin et al., 2001). In Table 1, we compare, for various noise levels $\sigma$, our model with the simplified DRUNet (called "DRUNet *light*") that has the same architecture as our GS-DRUNet (2 residual blocks) and that is trained (with $L^2$ loss) without the conservative field constraint. We also provide comparisons with the original DRUNet (Zhang et al., 2021) (with 4 residual blocks at each scale and trained with $L^1$ loss) and two state-of-the-art denoisers encountered in the PnP literature: FFDNet (Zhang et al., 2018) and DnCNN (Zhang et al., 2017a). For each network, we indicate in Table 1 the average runtime while processing a $256 \times 256$ color image on one Tesla P100 GPU.

Our GS-DRUNet denoiser, despite being constrained to be an exact conservative field, reaches the performance of (and even slightly outperforms) its unconstrained counterpart DRUNet *light*. Second, departing from the latter, we are able to reduce the processing time by a large margin ($\div 7$) while keeping close PSNR to the original DRUNet (around -0.05dB) and maintaining a significant PSNR gap (around +0.5dB) with other deep denoisers like DnCNN and FFDNet. Note that the time difference between GS-DRUNet and DRUNet *light* is due to the computation of $\nabla g_\sigma$ via backpropagation. These results indicate that GS-DRUNet is likely to yield a competitive and fast PnP algorithm.

| $\sigma(./255)$ | 5 | 15 | 25 | 50 | Time (ms) |
|---|---|---|---|---|---|
| FFDNet | 39.95 | 33.53 | 30.84 | 27.54 | 1.9 |
| DnCNN | 39.80 | 33.55 | 30.87 | 27.52 | 2.3 |
| DRUNet | **40.31** | **33.97** | **31.32** | **28.08** | 69.8 |
| DRUNet *light* | 40.19 | 33.89 | 31.25 | 28.00 | 6.3 |
| GS-DRUNet | 40.26 | 33.90 | 31.26 | 28.01 | 10.4 |

Table 1: Average PSNR denoising performance and runtime of our GS denoiser on $256 \times 256$ center-cropped images from the CBSD68 dataset, for various noise levels $\sigma$. While keeping small runtime, GS-DRUNet slightly outperforms its unconstrained counterpart DRUNet *light* and outdistances the deep denoisers FFDNet and DnCNN.

## 5.2   Plug-and-Play image restoration

We show in this section that, in addition to being convergent, our PnP Alg. 1 reaches state-of-the-art performance in deblurring and super-resolution (Sections 5.2.1 and 5.2.2) and realizes relevant inpainting (Appendix J.3). In all cases, we seek an estimate $x$ of a clean image $\hat{x} \in \mathbb{R}^n$, from an observation obtained as $y = A\hat{x} + \xi_\nu \in \mathbb{R}^m$, with $A$ a $m \times n$ degradation matrix and $\xi_\nu$ a white Gaussian noise with zero mean and standard deviation $\nu$. The objective function minimized by Alg. 1 is

$$F(x) = \frac{1}{2\nu^2}||Ax - y||^2 + \frac{\lambda}{2}||N_\sigma(x) - x||^2. \tag{12}$$

In practice, for $\nu > 0$, we multiply $F$ in equation (12) by $\nu^2$ and consider $F = f + \lambda_\nu g_\sigma$ with $f(x) = \frac{1}{2}||Ax - y||^2$, $g_\sigma(x) = \frac{1}{2}||N_\sigma(x) - x||^2$ and $\lambda_\nu = \lambda\nu^2$. With this formulation, the convergences of iterates and objective values are guaranteed by Theorems 1 and 2. We also demonstrate in Appendix J.3 that our framework can be extended to other kinds of objective functions. For example, inpainting noise-free input images leads to a non differentiable data-fidelity term $f$.

Due to the large computational time of some compared methods, we use for evaluation and comparison a subset of 10 color images taken from the CBSD68 dataset (CBSD10) together with the 3 famous set3C images (butterfly, leaves and starfish). Quantitative results run on the full CBSD68 dataset are given in Appendix J. All images are center-cropped to the size $256 \times 256$. For each IR problem, we provide default values for the parameters $\sigma$ and $\lambda$ that can be used to treat sucessfully a large class of images and degradations. The influence of both parameters is analyzed in Appendix J.5. Performance can be marginally improved by tuning $\lambda$ for each image, for example with the method of Wei et al. (2020) based on reinforcement learning. In our experiments, backtracking is performed with $\eta = 0.9$ and $\gamma = 0.1$. We observe (see Appendix B) that on a majority of images, the Lipschitz constant $L$ of $\nabla g_\sigma$ is slightly larger than 1. As convergence is ensured for $\lambda_\nu\tau = \nu^2\lambda\tau < \frac{1}{L}$, we set the initial stepsize to $\tau_0 = (\nu^2\lambda)^{-1}$. At the first iteration, the gradient step in equation (9) is thus exactly $D_\sigma$. In the majority of our experiments, backtracking is never activated. The algorithm is initialized with a proximal step and terminates when the relative difference between consecutive values of the objective function is less than $\epsilon$ or the number of iterations exceeds $K$.

### 5.2.1 DEBLURRING

For image deblurring, the degradation operator $A = H$ is a convolution performed with circular boundary conditions. Therefore, $H = \mathcal{F}^*\Lambda\mathcal{F}$, where $\mathcal{F}$ is the orthogonal matrix of the discrete Fourier transform (and $\mathcal{F}^*$ its inverse), and $\Lambda$ is a diagonal matrix. The proximal operator of the data-fidelity term $f(x) = \frac{1}{2}||Hx - y||^2$ involves only element-wise inversion and writes

$$\text{Prox}_{\tau f}(z) = \mathcal{F}^*(I_n + \tau\Lambda^*\Lambda)^{-1}\mathcal{F}(\tau H^T y + z). \tag{13}$$

We demonstrate the effectiveness of our method on a large variety of blur kernels (represented in Table 2) and noise levels. As in (Zhang et al., 2017b; Pesquet et al., 2021; Zhang et al., 2021), we use the 8 real-world camera shake kernels proposed in Levin et al. (2009) as well as the $9 \times 9$ uniform kernel and the $25 \times 25$ Gaussian kernel with standard deviation 1.6 (as in (Romano et al., 2017)). We consider Gaussian noise with 3 noise levels $\nu \in \{2.55, 7.65, 12.75\}/255$ *i.e.* $\nu \in \{0.01, 0.03, 0.05\}$. For all noise levels, we set $\sigma = 1.8\nu$, $\lambda_\nu = \nu^2\lambda = 0.1$ for motion blur (kernels (a) to (h)) and $\lambda_\nu = 0.075$ for static blur (kernels (i) and (j)). Initialization is done with $z_0 = y$ but we show in Appendix J.6 the robustness to the initialization. The stopping criteria are $\epsilon = 10^{-5}$ and $K = 400$.

We compare in Table 2 our method (GS-PnP) against the patch-based method EPLL (Zoran & Weiss, 2011; Hurault et al., 2018), the deep PnP methods IRCNN (Zhang et al., 2017b), DPIR (Zhang et al., 2021), MMO (Pesquet et al., 2021), and the "RED-FP" algorithm (Romano et al., 2017) (with TNRD denoiser (Chen & Pock, 2016)) referred to as RED. Both IRCNN and DPIR use PnP-HQS with a fast decrease of $\tau$ and $\sigma$ in a few iterations (8 iterations for DPIR) without guarantee of convergence. DPIR uses the DRUNet denoiser from Table 1. MMO is the only compared method that guarantees convergence by plugging a DnCNN denoiser trained with Lipschitz constraints (but the only given network was trained for very low noise level). Finally, as RED only treats the Y channel in the YCbCr color space, we also indicate in Appendix J.1, for RED and the proposed method, the PSNR evaluated on the Y channel only.

Among all methods, GS-PnP closely follows DPIR in terms of PSNR for low noise level but performs equally or better for higher noise levels. Other comparisons are conducted in Appendix J.1 on the Set3c and the full CBSD68 datasets (Tables 4 and 5). These results exhibit that GS-PnP reaches state-of-the-art in PnP deblurring for a variety of kernels and noise levels. We underline that the convergence of GS-PnP is guaranteed, whereas DPIR can asymptotically diverge (see Appendix J.7).

| $\nu$ | Method | (a) | (b) | (c) | (d) | (e) | (f) | (g) | (h) | (i) | (j) | Avg |
|---|---|---|---|---|---|---|---|---|---|---|---|---|
| 0.01 | EPLL | 28.32 | 28.24 | 28.36 | 25.80 | 29.61 | 27.15 | 26.90 | 26.69 | 25.84 | 26.49 | *27.34* |
| | RED | 30.47 | 30.01 | 30.29 | 28.09 | 31.22 | 28.92 | 28.90 | 28.67 | 26.66 | 28.45 | *29.17* |
| | IRCNN | 32.96 | 32.62 | 32.53 | 32.44 | 33.51 | 33.62 | 32.54 | 32.20 | 28.11 | 29.19 | *31.97* |
| | MMO | 32.35 | 32.06 | 32.24 | 31.67 | 31.77 | 33.17 | 32.30 | 31.80 | 27.81 | **29.26** | *31.44* |
| | DPIR | **33.76** | **33.30** | **33.04** | **33.09** | **34.10** | **34.34** | **33.06** | **32.77** | **28.34** | 29.16 | ***32.50*** |
| | GS-PnP | 33.52 | 33.07 | 32.91 | 32.83 | 34.07 | 34.25 | 32.96 | 32.54 | 28.11 | 29.03 | *32.33* |
| 0.03 | EPLL | 25.31 | 25.12 | 25.82 | 23.75 | 26.99 | 25.23 | 25.00 | 24.59 | 24.34 | 25.43 | *25.16* |
| | RED | 25.71 | 25.32 | 25.71 | 24.38 | 26.65 | 25.50 | 25.27 | 24.99 | 23.51 | 25.54 | *25.26* |
| | IRCNN | 28.96 | 28.65 | 28.90 | 28.38 | 30.03 | 29.87 | 28.92 | 28.52 | 25.92 | 27.64 | *28.58* |
| | IRCNN | 28.96 | 28.65 | 28.90 | 28.38 | 30.03 | 29.87 | 28.92 | 28.52 | 25.92 | 27.64 | *28.58* |
| | DPIR | **29.38** | **29.06** | **29.21** | **28.77** | 30.22 | **30.23** | **29.34** | 28.90 | 26.19 | 27.81 | ***28.91*** |
| | GS-PnP | 29.22 | 28.89 | 29.20 | 28.60 | **30.32** | 30.21 | 29.32 | **28.92** | **26.38** | **27.89** | *28.90* |
| 0.05 | EPLL | 24.08 | 23.91 | 24.78 | 22.57 | 25.68 | 23.98 | 23.70 | 23.19 | 23.75 | 24.78 | *24.04* |
| | RED | 22.78 | 22.54 | 23.13 | 21.92 | 23.78 | 22.97 | 22.89 | 22.67 | 22.01 | 23.78 | *22.84* |
| | IRCNN | 27.00 | 26.74 | 27.25 | 26.37 | 28.29 | 28.06 | 27.22 | 26.81 | 24.85 | 26.83 | *26.94* |
| | DPIR | **27.52** | **27.35** | **27.73** | **27.02** | 28.63 | **28.46** | 27.79 | 27.30 | 25.25 | 27.11 | *27.42* |
| | GS-PnP | 27.45 | 27.28 | 27.70 | 26.98 | **28.68** | 28.44 | **27.81** | **27.38** | **25.49** | **27.15** | ***27.44*** |

Table 2: PSNR(dB) comparison of image deblurring methods on CBSD10 with various blur kernels $k$ and noise levels $\nu$. Best and second best results are displayed in bold and underlined.

For qualitative comparison, we show in Figure 1(c-f) the deblurring obtained with various methods on the image "starfish" (from set3C). Note that our algorithm, compared to competing methods, can recover the sharpest edges. We also give convergence curves that empirically confirm the convergence of the values $F(x_k)$ (g) and of the residual $\min_{0 \leq i \leq k} ||x_{i+1} - x_i||^2/||x_0||^2$ (h). These observations are supported by the additional experiment shown in Appendix J.1, Figure 6.

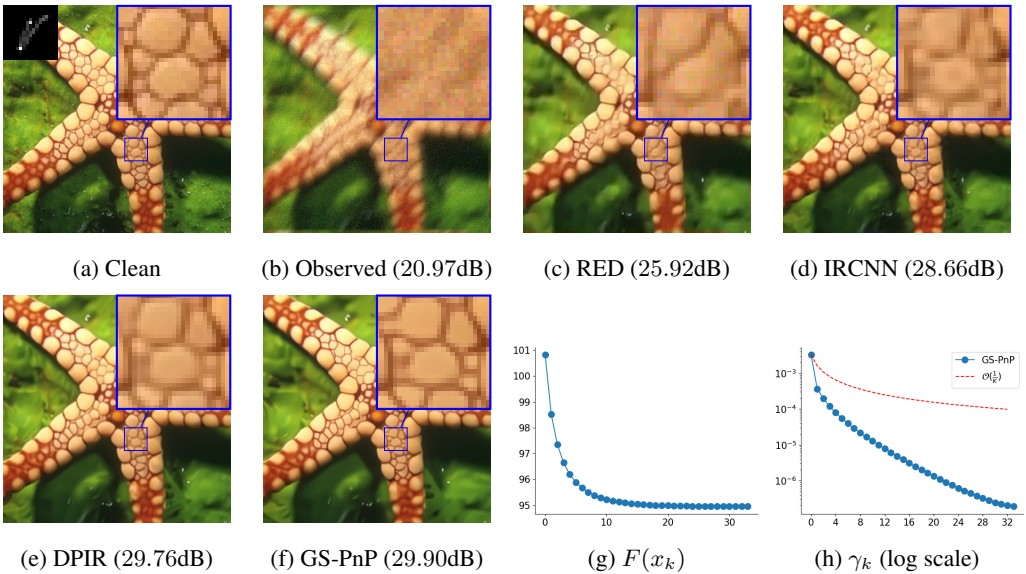

Figure 1: Deblurring with various methods of "starfish" degraded with the indicated blur kernel and input noise level $\nu = 0.03$. Note that our algorithm better recovers the structures. In (g) and (h), we show the evolution of $F(x_k)$ and $\gamma_k = \min_{0 \leq i \leq k} ||x_{i+1} - x_i||^2/||x_0||^2$ and in Appendix J.4 of the PSNR. We empirically verify convergence of functional values and residual. Note that the empirical convergence rate in (h) is faster than the $\mathcal{O}(\frac{1}{k})$ theoretical worst case rate established in Remark 2.

### 5.2.2 SUPER-RESOLUTION

For single image super-resolution, the low-resolution image $y \in \mathbb{R}^m$ is obtained from the high-resolution one $x \in \mathbb{R}^n$ via $y = SHx + \xi_\nu$ where $H \in \mathbb{R}^{n \times n}$ is the convolution with anti-aliasing kernel. The matrix $S$ is the standard s-fold downsampling matrix of size $m \times n$ and $n = s^2 \times m$. In this context, we make use of the closed-form calculation of the proximal map for the data-fidelity term $f(x) = \frac{1}{2}||SHx - y||^2$, given by Zhao et al. (2016):

$$\text{Prox}_{\tau f}(z) = \hat{z}_\tau - \frac{1}{s^2}\mathcal{F}^*\underline{\Lambda}^*\left(I_m + \frac{\tau}{s^2}\underline{\Lambda}\underline{\Lambda}^*\right)^{-1}\underline{\Lambda}\mathcal{F}\hat{z}_\tau, \tag{14}$$

where $\hat{z}_\tau = \tau H^T S^T y + z$ and $\underline{\Lambda} = [\Lambda_1, \ldots, \Lambda_{s^2}] \in \mathbb{R}^{m \times n}$, with $\Lambda = \text{diag}(\Lambda_1, \ldots, \Lambda_{s^2})$ a block-diagonal decomposition according to a $s \times s$ paving of the Fourier domain. Note that $I_m + \frac{\tau}{d}\underline{\Lambda}\underline{\Lambda}^*$ is a $m \times m$ diagonal matrix and its inverse is computed element-wise. As expected, with $s = 1$, equation (14) comes down to equation (13).

As in Zhang et al. (2021), we evaluate super-resolution performance on 8 Gaussian blur kernels represented in Table 3: 4 isotropic kernels with different standard deviations (0.7, 1.2, 1.6 and 2.0) and 4 anisotropic kernels. Results are averaged between isotropic and anisotropic. We consider downsampled images at scale $s = 2$ and $s = 3$ and Gaussian noise with 3 different noise levels $\nu \in \{0.01, 0.03, 0.05\}$. Our method (GS-PnP) is compared against bicubic upsampling, RED, IRCNN ("IRCNN+" from (Zhang et al., 2021)) and DPIR. We give again in Appendix J.2 the results obtained on the Set3C dataset (Table 7). All our results are obtained with $\lambda_\nu = \nu^2\lambda = 0.065$ and $\sigma = 2\nu$. Initialization $z_0$ is done with a bicubic interpolation of $y$ (with a shift correction (Zhang et al., 2021)) and the stopping criteria are $\epsilon = 10^{-6}$ and $K = 400$.

Besides being the only compared PnP method with convergence guarantee, GS-PnP outperforms in PSNR all other PnP algorithms over the considered range of blur kernels, noise levels and scale factors. We show in Figure 2 the super-resolution of the image "leaves" downsampled by 2, with an isotropic kernel and noise level $\nu = 0.03$. GS-PnP (f) recovers more accurately structures and color details than competing approaches (c-e), while converging in terms of function values (g) and residual (h). Additional visual comparisons are presented in Appendix J.2.

| Kernels | Method | $s = 2$ | | | $s = 3$ | | | $Avg$ |
|---|---|---|---|---|---|---|---|---|
| | | $\nu = 0.01$ | $\nu = 0.03$ | $\nu = 0.05$ | $\nu = 0.01$ | $\nu = 0.03$ | $\nu = 0.05$ | |
| | Bicubic | 24.85 | 23.96 | 22.79 | 23.14 | 22.52 | 21.62 | _23.15_ |
| | RED | 28.29 | 24.65 | 22.98 | 26.13 | 24.02 | 22.37 | _24.74_ |
| | IRCNN | 27.43 | 26.22 | 25.86 | 26.12 | 25.11 | 24.79 | _25.92_ |
| | DPIR | 28.62 | 27.30 | 26.47 | **26.88** | 25.96 | 25.22 | _26.74_ |
| | GS-PnP | **28.77** | **27.54** | **26.63** | 26.85 | **26.05** | **25.29** | _**26.86**_ |
| | Bicubic | 23.38 | 22.71 | 21.78 | 22.65 | 22.08 | 21.25 | _22.31_ |
| | RED | 26.33 | 23.91 | 22.45 | 25.38 | 23.40 | 21.91 | _23.90_ |
| | IRCNN | 25.83 | 24.89 | 24.59 | 25.36 | 24.36 | 23.95 | _24.83_ |
| | DPIR | **26.84** | 25.59 | 24.89 | **26.24** | 24.98 | **24.32** | _25.48_ |
| | GS-PnP | 26.80 | **25.73** | **25.03** | 26.18 | **25.08** | 24.31 | _**25.52**_ |

Table 3: PSNR(dB) comparison of image super-resolution methods on CBSD10 with various scales $s$, blur kernels $k$ and noise levels $\nu$. PNSR results are averaged over kernels at each row.

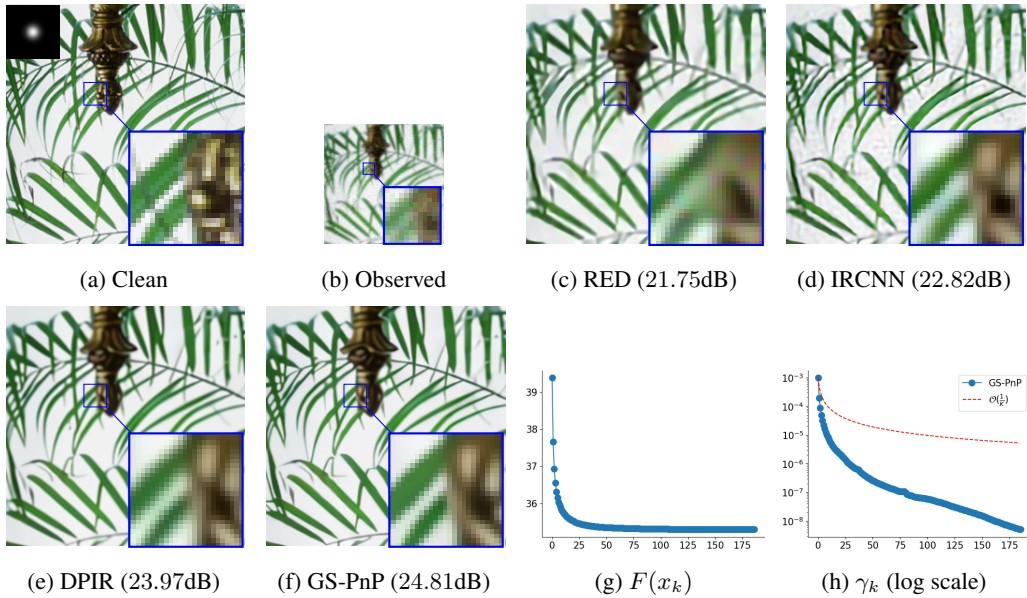

(a) Clean    (b) Observed    (c) RED (21.75dB)    (d) IRCNN (22.82dB)

(e) DPIR (23.97dB)    (f) GS-PnP (24.81dB)    (g) $F(x_k)$    (h) $\gamma_k$ (log scale)

Figure 2: Super-resolution with various methods on "leaves" (set3C) downsampled by 2, with the indicated blur kernel and input noise level $\nu = 0.03$. Note that our algorithm is the one that recovers sharpest leaves. In (g) and (h), we show the evolution of $F(x_k)$ and $\gamma_k = \min_{0 \le i \le k} ||x_{i+1} - x_i||^2 / ||x_0||^2$ and in AppendixJ.4 the evolution of the PSNR.. The empirical convergence rate is faster than the $\mathcal{O}(\frac{1}{k})$ theoretical worst case rate established in Remark 2.

## 6 CONCLUSION

In this work, we introduce a new PnP algorithm with convergence guarantees. A denoiser is trained to realize an exact gradient step on a regularization function that is formulated through a neural network. This denoiser is plugged in an iterative scheme closely related to PnP-HQS, which is proved to converge towards a stationary point of an explicit functional. One strength of this approach is to simultaneously allow for a non strongly convex (and non smooth) data-fidelity term with a denoiser that may not be nonexpansive. Experiments conducted on ill-posed imaging problems (deblurring, super-resolution, inpainting) confirm the convergence results and show that the proposed PnP algorithm reaches state-of-the-art image restoration performance. This work also opens several research perspectives. One could first examine which information is encoded in the proposed prior. For example, based on the sharp visual results, one can question if a relation can be drawn between this prior and the gradient energy or sparsity. Also, it would be interesting to see if the recovery analysis developed in (Liu et al., 2021) can adapt to the proposed framework.

# 7 REPRODUCIBILITY STATEMENT

Anonymous source code is given in supplementary material. It contains a README.md file that explains step by step how to run the algorithm and replicate the results of the paper. Moreover, the pseudocode of our algorithm is given Algorithm 1. In Section 5 it is precisely detailed how all the hyper-parameters are chosen and, for each experiment, which dataset is used. As for the theoretical results presented in Section 4, complete proofs are given in the appendixes.

# 8 ETHICS STATEMENT

We believe that this work does not raise potential ethical concerns.

## ACKNOWLEDGEMENTS

This work was funded by the French ministry of research through a CDSN grant of ENS Paris-Saclay. This study has also been carried out with financial support from the French Research Agency through the PostProdLEAP and Mistic projects (ANR-19-CE23-0027-01 and ANR-19-CE40-005).

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

## A    COMPARISON WITH THE PNP LITERATURE

### A.1    LIMITATIONS OF PREVIOUS PNP METHODS

In the existing literature, PnP approaches have one of the following limitations:

- They are not able to provide proof of convergence when non strongly convex data-fidelity terms are involved (Ryu et al., 2019), which is the case of some classical IR problems such as deblurring, super-resolution or inpainting.
- They are restricted to (nearly) nonexpansive denoisers (Reehorst & Schniter, 2018; Ryu et al., 2019; Sun et al., 2019b; Xu et al., 2020) or denoisers with a symmetric Jacobian (Romano et al., 2017). But it has already been shown that imposing symmetric Jacobian or Lipschitz property on a deep denoiser network alters its denoising performance (Bohra et al., 2021; Hertrich et al., 2021). We highlight that it was already empirically observed (Romano et al., 2017; Zhang et al., 2021) that the performance of the denoiser directly impacts the performance of the corresponding PnP scheme for IR.
- They show convergence of iterates thanks to decreasing time steps (Chan et al., 2016), but there is no characterization of the obtained solution (it is not a minima or a critical point of any functional).

On the other hand, our method is proved to converge to a stationary point of an explicit functional including a non strongly convex data-fidelity term. It also relies on a (possibly expansive) denoiser that, although being constrained to be a conservative vector field, allows to produce state-of-the-art results for various ill-posed IR problems.

### A.2    ON THE REGULARIZATION $g_\sigma$.

We first underline that the main point of our method is to define the denoiser as $D_\sigma = \mathrm{Id} - \nabla g_\sigma$. The choice for $g_\sigma$ is important for the denoising performance. With respect to the convergence properties of GS-PnP, this is nevertheless a secondary issue, as our method would converge for other differentiable regularizers $g_\sigma$.

The proposed regularization $g_\sigma(x) = \frac{1}{2}||x - N_\sigma(x)||^2$ was previously mentioned in the RED original paper (Romano et al., 2017) (but explicitly left aside) and used in the DAEP paper (Bigdeli & Zwicker, 2017). The main difference between our regularizer and the one alternately proposed in RED and DAEP is the following:

- RED and DAEP both consider a generic given pretrained denoiser $D_\sigma : \mathbb{R}^n \to \mathbb{R}^n$, which is then associated with the regularizer $g_\sigma(x) = \frac{1}{2}||x - D_\sigma(x)||^2$ and used as such in IR problems.
- In our method, we set $g_\sigma(x) = \frac{1}{2}||x - N_\sigma(x)||^2$ (with $N_\sigma : \mathbb{R}^n \to \mathbb{R}^n$ differentiable) and then we train the denoiser as $D_\sigma = \mathrm{Id} - \nabla g_\sigma$ with the loss function $||D_\sigma(x + \xi) - x||^2$ for clean images $x$ and additive white Gaussian noise (AWGN) $\xi$.

With this new formulation, we are ensured that $D_\sigma = \mathrm{Id} - \nabla g_\sigma$ is inherently a conservative vector field, without further assumptions on $N_\sigma$. Thanks to this relation, the (slightly modified) PnP-HQS given in relation (9) becomes a proximal gradient descent (PGD). We can then make use of convergence results of the PGD algorithm in the nonconvex setting to show the convergence of PnP-HQS.

In contrast to the original RED paper, we aimed at finding one setting of Plug-and-Play image restoration that allows for a convergence proof with sufficiently general hypotheses. For this purpose, we had to consider this very particular form of regularization.

### A.3 Recent literature on Regularization by denoising

We here provide a more detailed discussion on the follow-up literature on RED.

In parallel to the RED method (Romano et al., 2017), the authors of Bigdeli & Zwicker (2017) propose to use the regularization, mentioned but not exploited in RED, $g(x) = ||D_\sigma(x) - x||^2$, where $D_\sigma$ is a pretrained denoising autoencoder. Next, Bigdeli et al. (2017) extended this work with a new prior, which is the Gaussian-smoothed version of the natural image prior. Inspired by Tweedie's formula, they approximate the gradient of this log smoothed prior with the residual of a pretrained denoising autoencoder. With this new formulation, it is possible to optimize on the restored image but also on other parameters (e.g. the noise level and the used blur kernel).

Initially designed in the context of convex data-fidelity term, RED Romano et al. (2017) has been applied in the nonconvex setting for phase retrieval problems in prDeep Metzler et al. (2018).

Regularization by Artifact-Removal (RARE) (Liu et al., 2020) extends the RED framework by replacing the denoiser by a more general artifact-removal operator. The main advantage of this operator is that it can be trained without groundtruth data, but only by mapping pairs of artifact and noise contaminated images obtained directly from undersampled measurements. This is particularly useful for medical imaging applications where it is difficult to acquire fully-sampled training data.

The convergence of the original RED algorithm is discussed in Reehorst & Schniter (2018). The authors provide a convergence proof for RED-PGD which requires the denoiser to be nonexpansive, which, as detailed in the previous sections, is a restrictive hypothesis.

The authors of Liu et al. (2021) provide a recovery guarantee for the PnP framework, meaning convergence to a $x^*$ that satisfies $y = Ax^*$ while being in the set $\mathsf{Fix}(D)$ of the fixed points of $D$. More precisely, they show the convergence of the PnP-PGD method towards such a true solution $x^*$ under the assumptions that the denoiser residual $R = \mathrm{Id} - D$ is bounded and Lipschitz, and that the measurement operator satisfies a "set-restricted eigenvalue condition" (S-REC, which can be understood as strong convexity on the image of the denoiser $\mathsf{Im}(D)$). Under the additional assumptions that the denoiser is nonexpansive and that there exists $x \in \mathsf{Fix}(D)$ that is also critical for the regularizer $g$, they show that PnP and RED have the same solutions. As mentioned by the authors, it is nevertheless difficult to verify the S-REC condition for a given measurement operator: since $\mathsf{Im}(D)$ is not explicit, it is not clear how much S-REC relaxes the strong convexity. As explained in Sections 2 and 3, our results do not require strong convexity of the data-fidelity term.

Instead of including an explicit regularization in the functional, RED-PRO (Cohen et al., 2021) aims at minimizing the data-fidelity term on the set $\mathsf{Fix}(D)$ of fixed points of a generic denoiser $D$. The study is conducted under the hypothesis that the denoiser is demicontractive, which implies that $\mathsf{Fix}(D)$ is convex, thus leading to a convex optimization problem. However, this assumption seems difficult to verify in practice and the existence of fixed points for the RED-PRO operator does not appear straightforward. In contrast, the fixed points of the GS-PnP operator are directly related to the stationary points of the global functional $F = f + \lambda g_\sigma$ (Lemma 1 in Appendix C), whose existence is guaranteed as soon as $F$ is coercive (see the discussion in Appendix D).

ASYNC-RED (Sun et al., 2020) enables faster computation of RED by decomposing the inference into a sequence of partial (block-coordinate) updates on $x$ which can be executed asynchronously in parallel over a multicore system. As in their previous work BC-RED (Sun et al., 2019a), the authors propose to further reduce the computational time by using only a random subset of measurements at every iteration. Convergence of ASYNC-RED is shown, provided the denoiser is nonexpansive. A possible future extension of our work is the integration of the ASYNC framework to accelerate GS-PnP for large scale imaging inverse problems. As our GS-PnP converges without assuming nonexpansiveness of the denoising operation, it would be interesting to see if one can adapt the GS-PnP convergence properties to an ASYNC-GSPnP algorithm.

## B Lipschitz constant of $\nabla g_\sigma$

First, let us give a result which ensures that a large class of neural networks trained with differentiable activation functions have Lipschitz gradients with respect to the input image.

**Proposition 2.** *Let $H = h_p \circ \ldots \circ h_1$ be a composition of differentiable functions $h_i : \mathbb{R}^{d_{i-1}} \to \mathbb{R}^{d_i}$. Let us assume that for any $i$ the differential map $h'_i$ is bounded and Lipschitz. Then $H'$ is Lipschitz.*

*Proof.* Let us denote $H_i = h_i \circ \ldots \circ h_1$ (and by convention, $H_0 = \mathrm{Id}$). Let $\|h'_i\|_\infty$ be the best uniform bound on the operator norms $\|h'_i(x)\|, x \in \mathbb{R}^{d_{i-1}}$ (which is also the best Lipschitz constant of $h_i$). Let us also denote $\|h'_i\|_{\mathrm{Lip}}$ the Lipschitz constant of $h'_i$. The chain rule gives that for any $x$, $H'(x)$ can be expressed as a composition of linear maps

$$H'(x) = h'_p(H_{p-1}(x))h'_{p-1}(H_{p-2}(x))\ldots h'_1(x) \tag{15}$$

Therefore, for any $x, y$,

$$H'(x) - H'(y) = \sum_{i=0}^{p-1} h'_p(H_{p-1}(x))\ldots h'_{i+2}(H_{i+1}(x))h'_{i+1}(H_i(x))h'_i(H_{i-1}(y))\ldots h'_1(y) \tag{16}$$

$$- h'_p(H_{p-1}(x))\ldots h'_{i+2}(H_{i+1}(x))h'_{i+1}(H_i(y))h'_i(H_{i-1}(y))\ldots h'_1(y). \tag{17}$$

We can thus bound the operator norms

$$\|H'(x) - H'(y)\| \le \sum_{i=0}^{p-1} \Big( \|h'_p(H_{p-1}(x))\ldots h'_{i+2}(H_{i+1}(x))\| \tag{18}$$

$$\|h'_{i+1}(H_i(x)) - h'_{i+1}(H_i(y))\|\|h'_i(H_{i-1}(y))\ldots h'_1(y)\| \Big). \tag{19}$$

and thus

$$\|H'(x) - H'(y)\| \le \sum_{i=0}^{p-1} \Big( \prod_{j \ne i+1} \|h'_j\|_\infty \Big) \|h'_{i+1}\|_{\mathrm{Lip}} \|H'_i\|_\infty \|x - y\| \tag{20}$$

which concludes because the chain-rule ensures that $\|H'_i\|_\infty \le \|h'_i\|_\infty \ldots \|h'_1\|_\infty$. □

Proposition 2 applies for a neural network obtained as a composition of fully-connected layers with ELU activation functions, that is, by composing functions of the form

$$h(x) = E(Ax + b) \tag{21}$$

where $A$ is a matrix, $b$ a vector and $E$ is the element-wise ELU defined by

$$E(x)_i = \begin{cases} x_i & \text{if } x_i \ge 0 \\ e^{x_i} - 1 & \text{if } x_i < 0 . \end{cases} \tag{22}$$

It is easy to see that $E$ is differentiable and that $E'$ is 1-Lipschitz with $\|E'\|_\infty \le 1$. Therefore

$$h'(x) = E'(Ax + b)A \tag{23}$$

is also bounded and Lipschitz.

Let us also mention that this proposition encompasses the case of U-nets which, in addition to composing fully-connected layers, also integrates skip-connections. For example, taking a skip-connection on a composition $h_3 \circ h_2 \circ h_1$ amounts to defining

$$H(x) = h_3\big( h_2(h_1(x)) , h_1(x) \big). \tag{24}$$

This can be simply rewritten $H = h_3 \circ \tilde{h}_2 \circ h_1$ where

$$\tilde{h}_2(x) = \big( h_2(h_1(x)), h_1(x) \big). \tag{25}$$

It is then clear that $\tilde{h}_2$ has bounded Lipschitz differential as soon as $h_1$ and $h_2$ do.

The bound obtained in the proof of Proposition 2 is exponential in the depth of the neural network. We now provide some experiments showing that, in practice, the Lipschitz constant of $\nabla g_\sigma$ does not explode. We show in Figure 3, for various noise levels $\sigma$, the distribution of the spectral norms $\|\nabla^2 g_\sigma(x)\|_S$ on the training image set $X$, estimated with power iterations. The computed value varies a lot across images. Hence approximating the Lipschitz constant of $\nabla g_\sigma$ with $L = \max_{x \in X} \|\nabla^2 g_\sigma(x)\|_S$ would lead to under-estimated stepsizes and slow convergence on most images. Backtracking solves this issue by finding at each iteration the optimal stepsize allowing sufficient decrease of the objective function.

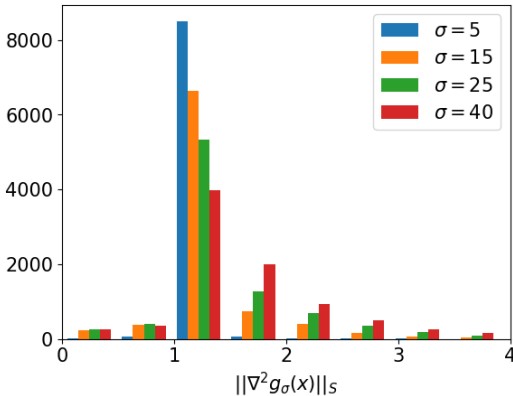

Figure 3: Histogram of the values of the spectral norm $||\nabla^2 g_\sigma(x)||_S$ evaluated on $128 \times 128$ images from the training dataset, degraded with white Gaussian noise with various standard deviations $\sigma$ (./255). Figure best seen in color.

## C  PROOF OF THEOREM 1

We first remind that a function $f : \mathbb{R}^n \longrightarrow \mathbb{R} \cup +\infty$ is proper if its domain

$$dom(f) = \{x \in \mathbb{R}, f(x) < +\infty\} \tag{26}$$

is non empty. Also, recall that $f$ is lower semicontinuous at $x^*$ if $\liminf_{x \to x^*} f(x) \geq f(x^*)$.

*Proof.*

(i) For ease of notation, we consider $\lambda = 1$. The generalisation for any $\lambda > 0$ is straightforward by rescaling $g_\sigma$ (and $L$) accordingly. We denote the proximal gradient fixed point operator $T_\tau = \text{Prox}_{\tau f} \circ (\text{Id} - \tau \nabla_x g_\sigma)$, the objective function $F = f + g_\sigma$ and we introduce

$$Q_\tau(x, y) = g_\sigma(y) + \langle x - y, \nabla g_\sigma(y) \rangle + \frac{1}{2\tau} ||x - y||^2 + f(x). \tag{27}$$

We have

$$\begin{aligned}
\arg \min_x Q_\tau(x, y) &= \arg \min_x g_\sigma(y) + \langle x - y, \nabla g_\sigma(y) \rangle + \frac{1}{2\tau} ||x - y||^2 + f(x) \\
&= \arg \min_x f(x) + \frac{1}{2\tau} ||x - (y - \tau \nabla g_\sigma(y))||^2 \\
&= \text{Prox}_{\tau f} \circ (\text{Id} - \tau \nabla_x g_\sigma)(y) = T_\tau(y).
\end{aligned} \tag{28}$$

By definition for the $\arg \min$, $x_{k+1} = T_\tau(x_k) \Rightarrow Q_\tau(x_{k+1}, x_k) \leq Q_\tau(x_k, x_k)$. Moreover, with $g_\sigma$ being $L$-smooth, we have by the descent lemma, for any $\tau \leq \frac{1}{L}$ and any $x, y \in \mathbb{R}^n$,

$$g_\sigma(x) \leq g_\sigma(y) + \langle x - y, \nabla g_\sigma(y) \rangle + \frac{1}{2\tau} ||x - y||^2, \tag{29}$$

so that for every $x, y \in \mathbb{R}^n$,

$$Q_\tau(x, x) = F(x) \quad \text{and} \quad Q_\tau(x, y) \geq F(x). \tag{30}$$

Therefore, at iteration $k$,

$$F(x_{k+1}) \leq Q_\tau(x_{k+1}, x_k) \leq Q_\tau(x_k, x_k) = F(x_k). \tag{31}$$

$(F(x_k))$ is thus non-increasing. Since $F$ is lower-bounded, $(F(x_k))$ thus converges to a limit $F^*$.

(ii) Note that $Q_\tau(x_{k+1}, x_k) \leq Q_\tau(x_k, x_k)$ implies

$$f(x_{k+1}) \leq f(x_k) - \langle x_{k+1} - x_k, \nabla g_\sigma(x_k) \rangle - \frac{1}{2\tau} ||x_{k+1} - x_k||^2. \tag{32}$$

Using also (29) with stepsize $\frac{1}{L}$, we get

$$
\begin{aligned}
F(x_{k+1}) &= f(x_{k+1}) + g_\sigma(x_{k+1}) \\
&\leq f(x_k) - \langle x_{k+1} - x_k, \nabla g_\sigma(x_k) \rangle - \frac{1}{2\tau}||x_{k+1} - x_k||^2 \\
&\quad + g_\sigma(x_k) + \langle x_{k+1} - x_k, \nabla g_\sigma(x_k) \rangle + \frac{L}{2}||x_{k+1} - x_k||^2 \\
&= F(x_k) - \left( \frac{1}{2\tau} - \frac{L}{2} \right) ||x_{k+1} - x_k||^2.
\end{aligned}
\tag{33}
$$

Summing over $k = 0, 1, ..., m$ gives

$$
\begin{aligned}
\sum_{k=0}^{m} ||x_{k+1} - x_k||^2 &\leq \frac{1}{\frac{1}{2\tau} - \frac{L}{2}} \left( F(x_0) - F(x_{m+1}) \right) \\
&\leq \frac{1}{\frac{1}{2\tau} - \frac{L}{2}} \left( F(x_0) - F^* \right).
\end{aligned}
\tag{34}
$$

Therefore, $\lim_{k \to \infty} ||x_{k+1} - x_k|| = 0$.

(iii) We begin by the two following lemmas characterizing the proximal gradient descent operator $T_\tau = \mathrm{Prox}_{\tau f} \circ (\mathrm{Id} - \tau \nabla_x g_\sigma)$.

**Lemma 1.** *With the assumptions of Theorem 1, for $x^* \in \mathbb{R}^n$, $x^*$ is a fixed point of the proximal gradient descent operator $T_\tau = \mathrm{Prox}_{\tau f} \circ (\mathrm{Id} - \tau \nabla_x g_\sigma)$, i.e. $T_\tau(x^*) = x^*$, if and only if $x^*$ is a stationary point of problem (10), i.e. $-\nabla g_\sigma(x^*) \in \partial f(x^*)$.*

*Proof.* By definition of the proximal operator, we have

$$
\begin{aligned}
T_\tau(x^*) = x^* &\Leftrightarrow x^* = \mathrm{Prox}_{\tau f} \circ (\mathrm{Id} - \tau \nabla_x g_\sigma)(x^*) \\
&\Leftrightarrow x^* - \tau \nabla_x g_\sigma(x^*) - x^* \in \tau \partial f(x^*) \\
&\Leftrightarrow -\nabla_x g_\sigma(x^*) \in \partial f(x^*).
\end{aligned}
\tag{35}
$$

$\square$

**Lemma 2.** *With the assumptions of Theorem 1, $T_\tau$ is $1 + \tau L$ Lipschitz.*

*Proof.* Using the fact that for $f$ convex, $\mathrm{Prox}_{\tau f}$ is 1-Lipschitz (Bauschke & Combettes, 2011, Proposition 12.28), and by the Lipschitz property of $\nabla_x g_\sigma$,

$$
\begin{aligned}
||T_\tau(x) - T_\tau(y)|| &= ||\mathrm{Prox}_{\tau f} \circ (\mathrm{Id} - \tau \nabla_x g_\sigma)(x) - \mathrm{Prox}_{\tau f} \circ (\mathrm{Id} - \tau \nabla_x g_\sigma)(y)|| \\
&\leq ||(\mathrm{Id} - \tau \nabla_x g_\sigma)(x) - (\mathrm{Id} - \tau \nabla_x g_\sigma)(y)|| \\
&\leq (1 + \tau L)||x - y||.
\end{aligned}
\tag{36}
$$

$\square$

Note that, by nonconvexity of $g_\sigma$, the fixed point operator $T_\tau$ is not necessarily nonexpansive, but we can still show the convergence of the fixed-point algorithm towards a critical point of the objective function. We can now turn to the proof of (iii). Let $x^*$ be a cluster point of $(x_k)_{k \geq 0}$. Then there exists a subsequence $(x_{k_j})_{j \geq 0}$ converging to $x^*$. We have $\forall j \geq 0$,

$$
\begin{aligned}
||x^* - T_\tau(x^*)|| &\leq ||x^* - x_{k_j}|| + ||x_{k_j} - T_\tau(x_{k_j})|| + ||T_\tau(x_{k_j}) - T_\tau(x^*)|| \\
&\leq (2 + \tau L)||x^* - x_{k_j}|| + ||x_{k_j} - T_\tau(x_{k_j})|| \text{ by Lemma 2.}
\end{aligned}
\tag{37}
$$

Using (ii), the right-hand side of the inequality tends to 0 as $j \to \infty$. Thus $||x^* - T_\tau(x^*)|| = 0$ and $x^* = T_\tau(x^*)$, which by Lemma 1 means that $x^*$ is a stationary point of problem (10). $\square$

# D   ON THE BOUNDEDNESS OF $(x_k)$

In order to obtain convergence of the iterates, in Theorem 2 the generated sequence $(x_k)$ is assumed to be bounded. In the experiments (Section 5), we observed that under the rest of assumptions of Theorem 2, boundedness was always verified. A sufficient condition for the boundedness of the iterates is the coercivity of the objective function, that is, $\lim_{|x| \to \infty} F(x) = +\infty$ (because the non-increasing property gives $F(x_k) \leq F(x_0)$).

Similar to Laumont et al. (2021), we can constrain $F$ to be coercive by choosing a convex compact set $C \subset \mathbb{R}^n$ where the iterates should stay and by adding an extra term to the regularization $g_\sigma$:

$$\hat{g}_\sigma(x) = g_\sigma(x) + \frac{1}{2}||x - \Pi_C(x)||^2 = \frac{1}{2}||x - N_\sigma(x)||^2 + \frac{1}{2}||x - \Pi_C(x)||^2 \tag{38}$$

with $\Pi_C$ the Euclidian projection on $C$. As $g_\sigma$ is differentiable, the gradient step becomes

$$(\text{Id} - \tau\lambda\nabla_x \hat{g}_\sigma)(x) = (\text{Id} - \tau\lambda\nabla_x g_\sigma) + \tau\lambda(x - \Pi_C(x)). \tag{39}$$

In our experiments, we choose the compact set $C$ as $C = [-1, 2]^n$. In practice we observe that all the iterates always remain in $C$ and that the extra regularization term $||x - \Pi_C(x)||^2$ is never activated. Therefore, we don't present this technical adaptation in Algorithm 1 but we let the reader aware that boundedness of $(x_k)$ is not a limiting assumption.

# E   KL PROPERTY

**Definition 1.** *Kurdyka-Lojasiewicz (KL) property (taken from Attouch et al. (2010))*

(a) *A function $f : \mathbb{R}^n \longrightarrow \mathbb{R} \cup +\infty$ is said to have the Kurdyka-Lojasiewicz property at $x^* \in dom(f)$ if there exists $\eta \in (0, +\infty)$, a neighborhood $U$ of $x^*$ and a continuous concave function $\psi : [0, \eta) \longrightarrow \mathbb{R}_+$ such that $\psi(0) = 0$, $\psi$ is $\mathcal{C}^1$ on $(0, \eta)$, $\psi' > 0$ on $(0, \eta)$ and $\forall x \in U \cap [f(x^*) < f < f(x^*) + \eta]$, the Kurdyka-Lojasiewicz inequality holds:*

$$\psi'(f(x) - f(x^*))dist(0, \partial f(x)) \geq 1. \tag{40}$$

(b) *Proper lower semicontinuous functions which satisfy the Kurdyka-Lojasiewicz inequality at each point of $dom(\partial f)$ are called KL functions.*

This condition can be interpreted as the fact that, up to a reparameterization, the function is sharp *i.e.* we can bound its subgradients away from 0. A big class of functions that have the KL-property is given by real semi-algebraic functions. For more details and interpretations, we refer to Attouch et al. (2010) and Bolte et al. (2010).

# F   ON THE ASSUMPTIONS OF THEOREMS 1 AND 2

In this section, we explicitly list and comment all the assumptions required by Theorems 1 and 2. These assumptions are standard in nonconvex optimization. We now detail why each assumption is verified for our plug-and-play image restoration algorithm.

**Assumptions of Theorem 1**:

- *Data-fidelity term $f : \mathbb{R}^n \to \mathbb{R} \cup \{+\infty\}$ proper lower semicontinous and convex.* This is a general assumption that includes most of the data-fidelity terms classically used in IR problems. Note that we do not require differentiability of $f$. Degradations with Gaussian, Poisson or Laplacian noise models fall into this hypothesis. As noticed in Remark 3, our results can even be easily extended to a nonconvex data-fidelity term $f$, which encompasses applications like phase retrieval Metzler et al. (2018). In practice, it is helpful to have $f$ proximable, *i.e.* $\text{Prox}_f$ with closed-form formula. Otherwise, $\text{Prox}_f$ needs to be calculated at each iteration with an optimization algorithm.

- *Regularization function $g_\sigma : \mathbb{R}^n \to \mathbb{R}$ proper lower semicontinous and differentiable with L-Lipschitz gradient.* We parametrize as $g_\sigma(x) = \frac{1}{2}||x - N_\sigma(x)||^2$ with $N_\sigma$ a differentiable neural network. This assumption on $g_\sigma$ is thus reasonable from a practical perspective. Indeed, using a network $N_\sigma$ with differentiable activation functions, our function $g_\sigma$ is differentiable with Lipschitz gradient (details and proof are given in Appendix B).

- *Functional $F = f + \lambda g_\sigma$ bounded from below.* This is straightforward as all the terms are positive.

- *The stepsize $\tau < \frac{1}{\lambda L}$.* This is handled by backtracking (see Section 4.2).

**Assumptions of Theorem 2**:

- *Assumptions of Theorem 1*

- *F verify the KL property.* The KL property (defined in Appendix E) has been widely used to study the convergence of optimization algorithms in the nonconvex setting (Attouch et al., 2010; 2013; Ochs et al., 2014). Very large classes of functions, in particular all the semi-algebraic functions, satisfy this technical property. It encompasses all the data-fidelity and regularization terms encountered in inverse problems.

- *The sequence $(x_k)$ given by the iterative scheme (9) is bounded.* As discussed in Appendix D, the boundedness can be ensured with a potential additional projection at each iteration. This is just a theoretical guarantee, as we observed that such a projection is never activated in practice.

## G  BACKTRACKING AND PROOF OF PROPOSITION 1

Before giving the proof of Proposition 1, we first point out that our backtracking line search is a classical Armijo-type backtracking strategy, already used for nonconvex optimization in (Beck, 2017, Chapter 10) or Ochs et al. (2014). Other procedures could be investigated in future work. For instance, Li & Lin (2015) uses a Barzilai-Borwein rule to initialize the backtracking line search. Scheinberg et al. (2014) and Calatroni & Chambolle (2019) have also proposed a backtracking strategy that allows for both decreasing and increasing of the stepsize.

We now give the proof of Proposition 1.

*Proof.* For a given stepsize $\tau$, we showed in Appendix C, equation (33) that

$$F(x_k) - F(T_\tau(x_k)) \geq \frac{1}{2}\left(\frac{1}{\tau} - L\right)||T_\tau(x_k) - x_k||^2. \tag{41}$$

Taking $\tau < \frac{1-2\gamma}{L}$, we get $\frac{1}{2}(\frac{1}{\tau} - L) > \frac{\gamma}{\tau}$ so that

$$F(x_k) - F(T_\tau(x_k)) > \frac{\gamma}{\tau}||T_\tau(x_k) - x_k||^2. \tag{42}$$

Hence, when $\tau < \frac{1-2\gamma}{L}$, the sufficient decrease condition equation (42) is satisfied and the backtracking procedure ($\tau \longleftarrow \eta\tau$) must end.

In the proof of Theorem 1, we can replace the sufficient decrease (33) by (42) and finish the proof with the same arguments. In the same way, in the proof of Theorem 2 given in (Attouch et al., 2013, Theorem 5.1), our sufficient decrease (42) replaces (Attouch et al., 2013, Equation (52)).

$\square$

## H    DRUNET *light* ARCHITECTURE

The architecture of the DRUNet *light* denoiser of (Zhang et al. (2021)) is given in Figure 4.

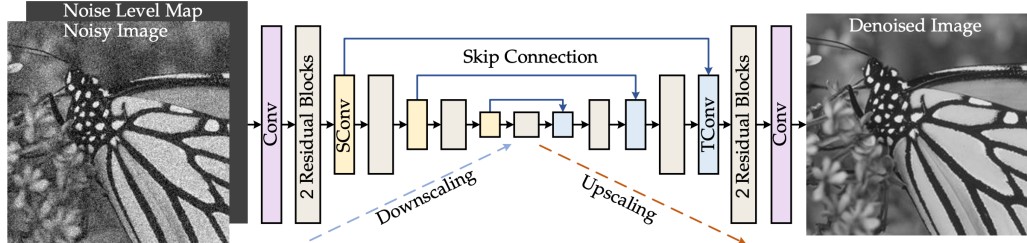

Figure 4: Architecture of the DRUNet *light* denoiser (Zhang et al. (2021)) used to parameterize $N_\sigma$.

## I    EXPANSIVENESS OF THE DENOISER

As $g_\sigma$ is not necessarily convex, our GS-DRUNet denoiser $D_\sigma = \text{Id} -\nabla g_\sigma$ is not necessarily non-expansive and neither is the gradient step $\text{Id} -\lambda\tau\nabla g_\sigma$. This is not an issue as, unlike previous theoretical PnP studies (Terris et al., 2020; Reehorst & Schniter, 2018), our convergence results do not require a nonexpansive denoising step. To advocate that our method converges without this assumption, we show in Figure 5 the evolution of $\frac{||D_\sigma(x_{k+1})-D_\sigma(x_k)||}{||x_{k+1}-x_k||}$ along the algorithm that was run to obtain the super-resolution results of Figure 7. In this experiment, backtracking did not get activated and stayed fixed at $\lambda\tau = 1$. The gradient step in the PGD algorithm was thus simply a denoising step $D_\sigma = \text{Id} -\lambda\tau\nabla g_\sigma$. Note that the Lipschitz constant of $D_\sigma$ goes above 1 but convergence is still observed as shown by the two convergence curves in Figure 7.

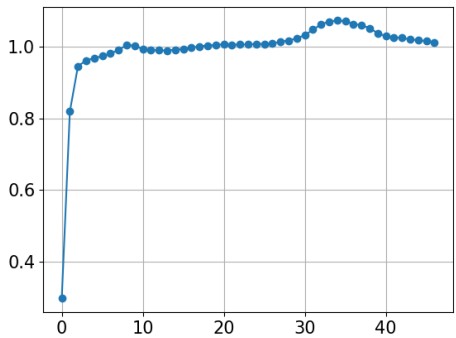

Figure 5: Lipschitz constant of $D_\sigma$ along the iterates of the algorithm when performing the super-resolution experiments presented Figure 7. Note that the Lipschitz constant goes above 1 *i.e.* $D_\sigma$ is not nonexpansive, but we still empirically verified convergence (see convergence curves Figure 7).

## J    ADDITIONAL EXPERIMENTS

### J.1    DEBLURRING

We give here additional image deblurring experiments. We first present the PSNR performance comparison on the Set3c dataset in Table 4. We also provide an evaluation of the 3 best methods (GS-PnP, DPIR and IRCNN) on the full CBSD68 dataset in Table 5. For fair comparison with RED, we also display in Table 6 the PSNR calculated on the Y channel only. An additional visual

comparison is finally shown in Figure 6. Details and comments are given in the corresponding captions.

| ν | Method | (a) | (b) | (c) | (d) | (e) | (f) | (g) | (h) | (i) | (j) | Avg |
|---|--------|-----|-----|-----|-----|-----|-----|-----|-----|-----|-----|-----|
| 0.01 | EPLL | 23.83 | 24.14 | 24.83 | 19.85 | 26.08 | 21.77 | 21.53 | 21.57 | 22.43 | 21.36 | *22.74* |
| | RED | 29.21 | 28.58 | 29.52 | 24.54 | 30.45 | 25.34 | 26.06 | 26.07 | 25.11 | 28.50 | *27.34* |
| | IRCNN | 33.36 | 33.06 | 33.11 | 32.87 | 34.24 | 34.08 | 33.25 | 32.87 | 27.78 | 29.67 | *32.45* |
| | MMO | 32.84 | 32.29 | 32.76 | 31.85 | 34.08 | 33.76 | 33.11 | 32.38 | 26.31 | 29.91 | *31.93* |
| | DPIR | **34.94** | **34.46** | **34.25** | **34.34** | **35.57** | **35.53** | **34.49** | **34.21** | 28.14 | 29.63 | ***33.56*** |
| | GS-PnP | 34.58 | 34.13 | 34.04 | 33.93 | 35.45 | 35.25 | 34.30 | 33.97 | **28.16** | **29.78** | *33.34* |
| 0.03 | EPLL | 21.21 | 21.10 | 22.65 | 18.78 | 24.12 | 20.77 | 20.42 | 19.89 | 20.61 | 20.60 | *21.02* |
| | RED | 25.42 | 24.89 | 25.69 | 22.67 | 26.86 | 23.84 | 24.06 | 23.87 | 21.49 | 25.45 | *24.43* |
| | IRCNN | 29.08 | 28.62 | 29.03 | 28.46 | 30.51 | 30.06 | 29.23 | 28.74 | 24.39 | 27.39 | *28.55* |
| | DPIR | **30.33** | 29.74 | 29.87 | **29.67** | 31.27 | 31.08 | 30.21 | 29.72 | 25.02 | 27.84 | *29.48* |
| | GS-PnP | 30.29 | **29.84** | **30.14** | 29.58 | **31.53** | **31.24** | **30.41** | **29.96** | **26.13** | **28.56** | ***29.77*** |
| 0.05 | EPLL | 19.84 | 19.60 | 21.40 | 17.71 | 22.77 | 19.68 | 19.02 | 18.24 | 19.81 | 20.12 | *19.82* |
| | RED | 21.93 | 21.27 | 22.79 | 20.32 | 24.01 | 22.05 | 22.06 | 21.41 | 19.79 | 23.21 | *21.88* |
| | IRCNN | 26.85 | 26.33 | 27.04 | 26.10 | 28.46 | 27.90 | 27.05 | 26.56 | 22.90 | 26.16 | *26.54* |
| | DPIR | 27.96 | 27.37 | 28.07 | 27.44 | 29.42 | 29.04 | 28.32 | 27.56 | 23.57 | 26.93 | *27.57* |
| | GS-PnP | **28.08** | **27.75** | **28.35** | **27.56** | **29.60** | **29.17** | **28.49** | **28.01** | **24.67** | **27.47** | ***27.91*** |

Table 4: PSNR(dB) comparison of image deblurring methods on set3C with various blur kernels $k$ and noise levels $\nu$. Best and second best results are displayed in bold and underlined. Similar to Table 2, for all kinds of kernels, the proposed method outperforms all competing methods at noise levels 0.03 and 0.05 and follows DPIR at lower noise level 0.01.

| ν | Method | (a) | (b) | (c) | (d) | (e) | (f) | (g) | (h) | (i) | (j) | Avg |
|---|--------|-----|-----|-----|-----|-----|-----|-----|-----|-----|-----|-----|
| 0.01 | IRCNN | 32.47 | 32.14 | 31.94 | 31.97 | 32.94 | 33.13 | 31.92 | 31.62 | 27.57 | **28.45** | *31.42* |
| | DPIR | **33.26** | **32.82** | **32.48** | **32.65** | **33.57** | **33.85** | **32.49** | **32.22** | 27.65 | 28.26 | ***31.93*** |
| | GS-PnP | 32.95 | 32.54 | 32.26 | 32.31 | 33.41 | 33.71 | 32.29 | 31.92 | 27.43 | 28.17 | *31.70* |
| 0.03 | IRCNN | 28.43 | 28.11 | 28.28 | 27.87 | 29.42 | 29.21 | 28.37 | 27.97 | 25.52 | 26.96 | *28.01* |
| | DPIR | **28.88** | **28.53** | **28.55** | **28.30** | 29.58 | **29.62** | **28.69** | 28.28 | 25.60 | 26.96 | ***28.30*** |
| | GS-PnP | 28.64 | 28.32 | **28.55** | 28.06 | **29.71** | 29.60 | **28.69** | **28.31** | 25.79 | 27.10 | *28.28* |
| 0.05 | IRCNN | 26.73 | 26.42 | 26.73 | 26.13 | 27.69 | 27.39 | 26.69 | 26.33 | 24.68 | 26.18 | *26.40* |
| | DPIR | **27.04** | **26.80** | **27.07** | **26.53** | 28.00 | 27.85 | 27.17 | **26.72** | 24.75 | 26.32 | *26.82* |
| | GS-PnP | 26.93 | 26.72 | **27.07** | 26.45 | **28.09** | **27.87** | **27.21** | **26.82** | 25.02 | 26.45 | ***26.86*** |

Table 5: PSNR(dB) performance of the fastest method (IRCNN/DPIR/GS-PnP) for image deblurring on the full CBSD68 dataset with various blur kernels $k$ and noise levels $\nu$, in the same conditions as Table 2. On CBSD10 (Table 2) or on CBSD68 (Table 5), we observed very similar performance gaps between the compared methods, which confirms that CBSD10 is large enough to compare accurately the PnP methods.

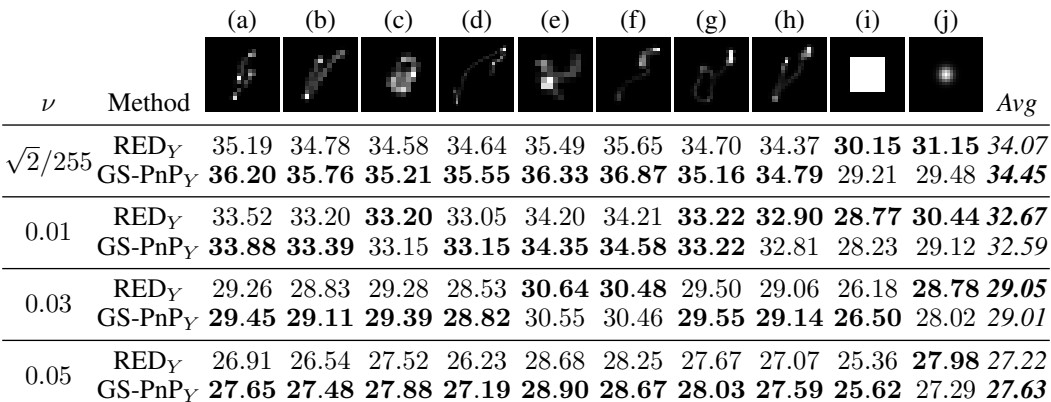

| $\nu$ | Method | (a) | (b) | (c) | (d) | (e) | (f) | (g) | (h) | (i) | (j) | Avg |
|---|---|---|---|---|---|---|---|---|---|---|---|---|
| $\sqrt{2}/255$ | RED$_Y$ | 35.19 | 34.78 | 34.58 | 34.64 | 35.49 | 35.65 | 34.70 | 34.37 | **30.15** | **31.15** | *34.07* |
| | GS-PnP$_Y$ | **36.20** | **35.76** | **35.21** | **35.55** | **36.33** | **36.87** | **35.16** | **34.79** | 29.21 | 29.48 | ***34.45*** |
| 0.01 | RED$_Y$ | 33.52 | 33.20 | **33.20** | 33.05 | 34.20 | 34.21 | **33.22** | **32.90** | **28.77** | **30.44** | ***32.67*** |
| | GS-PnP$_Y$ | **33.88** | **33.39** | 33.15 | **33.15** | **34.35** | **34.58** | **33.22** | 32.81 | 28.23 | 29.12 | *32.59* |
| 0.03 | RED$_Y$ | 29.26 | 28.83 | 29.28 | 28.53 | **30.64** | **30.48** | 29.50 | 29.06 | 26.18 | **28.78** | ***29.05*** |
| | GS-PnP$_Y$ | **29.45** | **29.11** | **29.39** | **28.82** | 30.55 | 30.46 | **29.55** | **29.14** | **26.50** | 28.02 | *29.01* |
| 0.05 | RED$_Y$ | 26.91 | 26.54 | 27.52 | 26.23 | 28.68 | 28.25 | 27.67 | 27.07 | 25.36 | **27.98** | 27.22 |
| | GS-PnP$_Y$ | **27.65** | **27.48** | **27.88** | **27.19** | **28.90** | **28.67** | **28.03** | **27.59** | **25.62** | 27.29 | ***27.63*** |

Table 6: PSNR(dB) performance, evaluated on the luminance channel in YcbCr color space, of RED and GS-PnP for image deblurring on CBSD10. Remind that our method treats the RGB image as a whole before being evaluated on the Y channel while RED treats the Y channel independently. Compared to Table 2, we add the case $\nu = \sqrt{2}/255$ as in RED original paper (Romano et al. (2017)). Note that RED was optimized for kernels (i) and (j) and $\nu = \sqrt{2}/255$, and outperforms our method in this set of conditions. However, over the variety of kernels and noise levels, and in particular for motion blur, our method generally outperforms RED.

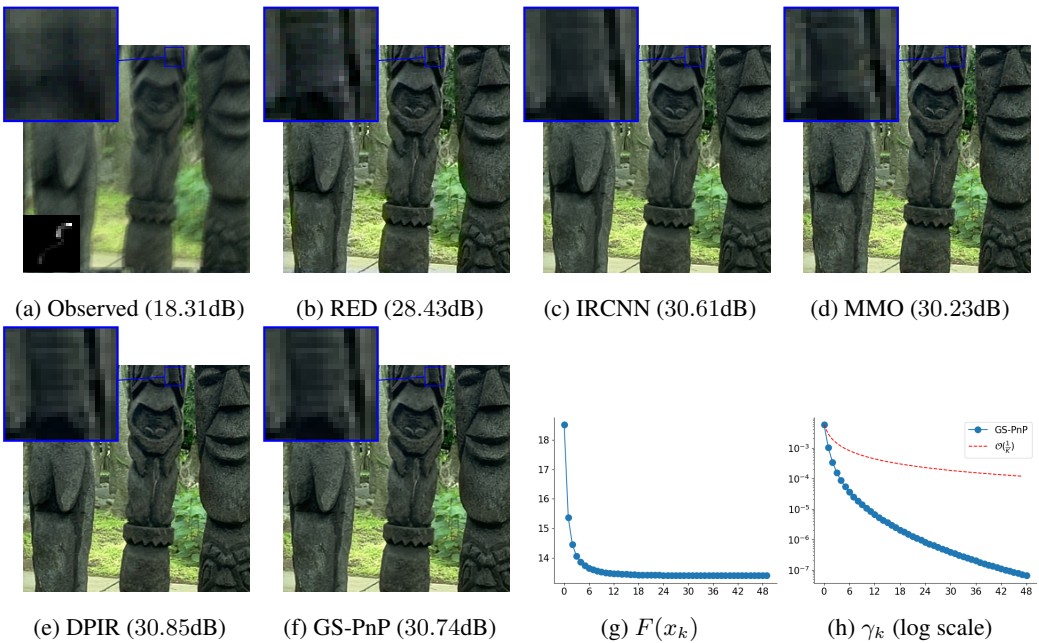

(a) Observed (18.31dB)  (b) RED (28.43dB)  (c) IRCNN (30.61dB)  (d) MMO (30.23dB)

(e) DPIR (30.85dB)  (f) GS-PnP (30.74dB)  (g) $F(x_k)$  (h) $\gamma_k$ (log scale)

Figure 6: Deblurring with various methods of an image from CSBD10 degraded with the indicated blur kernel and input noise level $\nu = 0.01$. In (g) and (h), we show the evolution of $F(x_k)$ and $\gamma_k = \min_{0 \le i \le k} ||x_{i+1} - x_i||^2 / ||x_0||^2$ along our algorithm. Note that GS-PnP and DPIR both recover fine textures while other methods tend to smooth details.

## J.2 SUPER-RESOLUTION

We also present additional super-resolution experiments. We realize a full PSNR performance comparison on the Set3c dataset Table 7. We show additional visual comparisons between methods Figure 7 and Figure 8. Details and comments are given in the corresponding captions.

| Kernels | Method | $s = 2$ | | | $s = 3$ | | | $Avg$ |
|---------|--------|---------------|---------------|---------------|---------------|---------------|---------------|-------|
| | | $\nu = 0.01$ | $\nu = 0.03$ | $\nu = 0.05$ | $\nu = 0.01$ | $\nu = 0.03$ | $\nu = 0.05$ | |
| | Bicubic | 21.92 | 21.54 | 20.90 | 19.76 | 19.53 | 19.11 | *20.46* |
| | RED | 28.22 | 25.62 | 23.61 | 24.91 | 23.38 | 21.82 | *24.59* |
| | IRCNN | 28.35 | 26.40 | 25.27 | 25.61 | 24.45 | 23.37 | *25.58* |
| | DPIR | 29.08 | 27.27 | 26.21 | **26.55** | 25.33 | 24.41 | *26.48* |
| | GS-PnP | **29.24** | **28.03** | **26.65** | 25.90 | **25.56** | **24.60** | ***27.00*** |
| | Bicubic | 19.82 | 19.58 | 19.16 | 18.95 | 18.76 | 18.40 | *19.11* |
| | RED | 24.72 | 22.55 | 21.10 | 22.82 | 21.64 | 20.19 | *22.17* |
| | IRCNN | 25.10 | 23.44 | 22.52 | 24.25 | 22.60 | 21.58 | *23.25* |
| | DPIR | **26.22** | 24.52 | 23.56 | **25.34** | 23.57 | 22.50 | ***24.29*** |
| | GS-PnP | 25.45 | **24.84** | **23.80** | 24.53 | **23.73** | **22.71** | *24.18* |

Table 7: PSNR(dB) comparison of image super-resolution methods on set3C with various scales $s$, blur kernels $k$ and noise levels $\nu$. Similar to Table 3, for isotropic and anisotropic kernels, the proposed method outperforms all competing methods at noise levels $0.03$ and $0.05$ and follows DPIR at lower noise level $0.01$.

| Kernels | Method | $s = 2$ | | | $s = 3$ | | | $Avg$ |
|---------|--------|---------------|---------------|---------------|---------------|---------------|---------------|-------|
| | | $\nu = 0.01$ | $\nu = 0.03$ | $\nu = 0.05$ | $\nu = 0.01$ | $\nu = 0.03$ | $\nu = 0.05$ | |
| | IRCNN | 26.97 | 25.86 | 25.45 | 25.60 | 24.72 | 24.38 | *25.50* |
| | DPIR | 27.79 | 26.58 | 25.83 | **26.05** | 25.27 | 24.66 | *26.03* |
| | GS-PnP | **27.88** | **26.81** | **26.01** | 25.97 | **25.35** | **24.74** | ***26.13*** |
| | IRCNN | 25.41 | 24.52 | 24.18 | 24.94 | 24.04 | 23.61 | *24.45* |
| | DPIR | **26.08** | 24.99 | 24.39 | **25.53** | 24.46 | 23.80 | *24.88* |
| | GS-PnP | 25.98 | **25.07** | **24.53** | 25.47 | **24.56** | **23.92** | ***24.92*** |

Table 8: PSNR(dB) performance of the fastest method (IRCNN/DPIR/GS-PnP) for image super-resolution on the full CBSD68 dataset with various blur kernels $k$ and noise levels $\nu$, in the same conditions as Table 3. Once again, on CBSD10 (Table 2) or on CBSD68 (Table 5), we observed very similar performance gaps between the compared methods, which again confirms that CBSD10 is large enough to compare accurately the PnP methods.

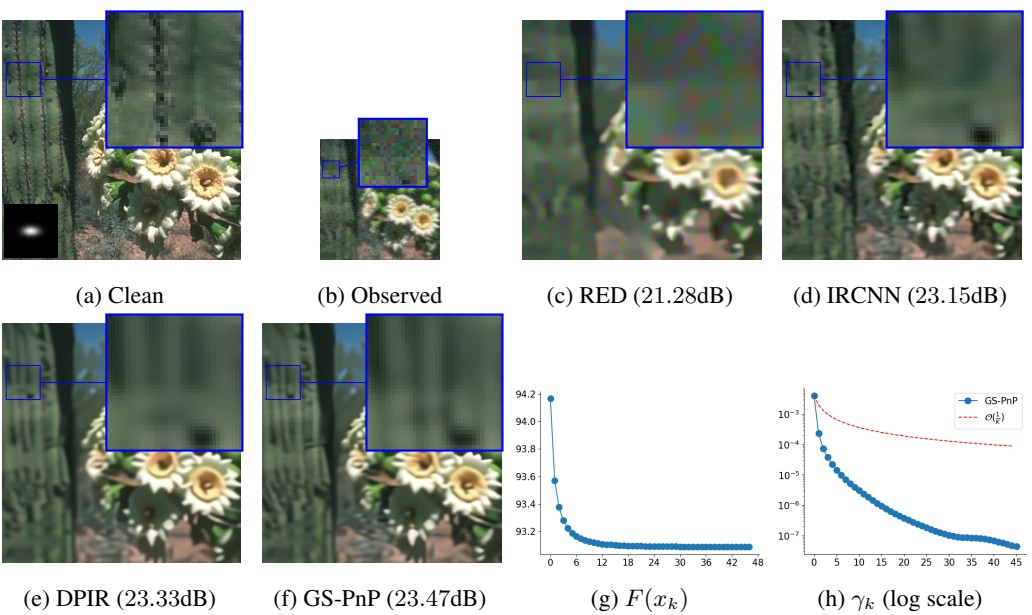

Figure 7: Super-resolution with various methods on a CBSD10 image degraded with the indicated blur kernel, $s = 2$ and input noise level $\nu = 0.05$. In (g) and (h), we show the evolution of $F(x_k)$ and $\gamma_k = \min_{0 \le i \le k} ||x_{i+1} - x_i||^2 / ||x_0||^2$ along our algorithm. One can notice that the proposed method GS-PnP manages to extract more structure in the zoomed area than the competing methods.

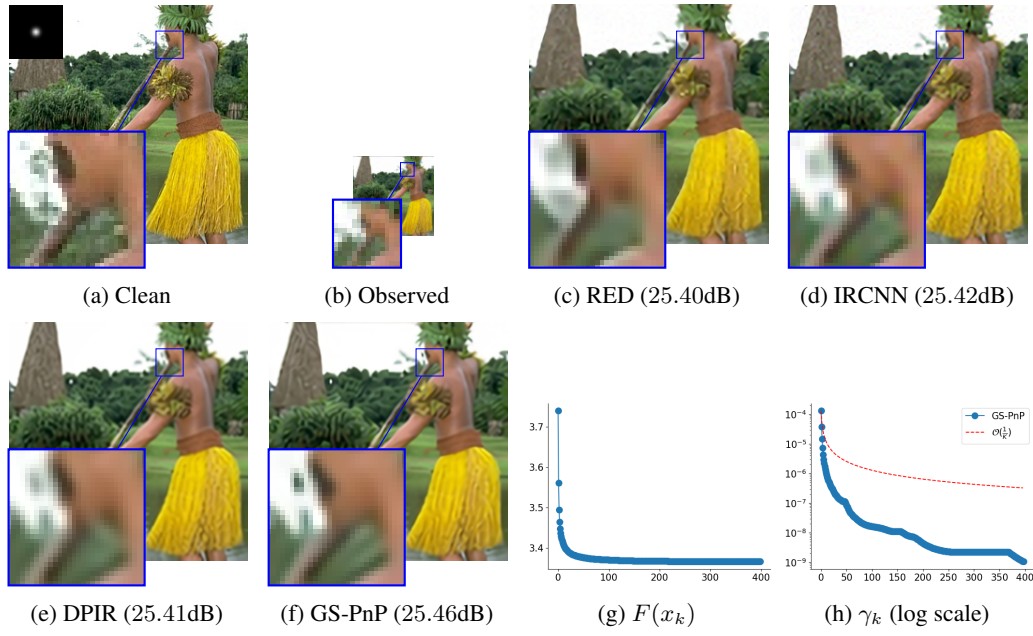

Figure 8: Super-resolution with various methods on a CBSD10 image degraded with the indicated blur kernel, $s = 3$ and input noise level $\nu = 0.01$. In (g) and (h), we show the evolution of $F(x_k)$ and $\gamma_k = \min_{0 \le i \le k} ||x_{i+1} - x_i||^2 / ||x_0||^2$ along our algorithm.

### J.3 INPAINTING (WITH NON-DIFFERENTIABLE DATA-FIDELITY TERM)

We now propose to apply our PnP scheme to image inpainting with the degradation model

$$y = Ax \tag{43}$$

where $A$ is a diagonal matrix with values in $\{0, 1\}$. For inpainting, no noise is added to the degraded image. In this context, the data-fidelity term is the indicator function of $A^{-1}(\{y\}) = \{x \mid Ax = y\}$: $f(x) = \iota_{A^{-1}(\{y\})}$ (which, by definition, equals 0 on $A^{-1}(\{y\})$ and $+\infty$ elsewhere). Despite being non differentiable, $f$ still verifies the assumptions of Theorems 1 and 2 and convergence is theoretically ensured. The proximal map becomes the orthogonal projection $\Pi_{A^{-1}(\{y\})}$

$$\text{Prox}_{\tau f}(x) = \Pi_{A^{-1}(\{y\})}(x) = Ay - Ax + x \tag{44}$$

In our experiments, the diagonal of $A$ is filled with Bernoulli random variables with parameter $p = 0.5$. We run our PnP algorithm with $\sigma = 10/255$. Given the form of $f$, we do not use the backtracking strategy and keep a fixed stepsize. Even if we do not exactly know the Lipschitz constant of $\nabla g_\sigma$, we observed in Figure 5 that, for small noise, it was almost always estimated as slightly larger than 1. We thus choose $\lambda\tau = 1$ and empirically confirm convergence with this choice in follow-up experiments (see Figure 9). The algorithm is initialized with $x_0 = y + 0.5(\text{Id} - A)y$ (masked pixels with value 0.5) and terminates when the number of iterations exceeds $K = 100$. We found it useful to run the first 10 iterations of the algorithm at larger noise level $\sigma = 50/255$. As $y$ does not have noise, we found preferable not to run the last extra gradient pass from Algorithm 1.

We show inpainting results on set3C images Figure 9. Our PnP restores the input images with high accuracy, including its small details. Furthermore, convergence of the residual at rate $\mathcal{O}(\frac{1}{k})$ is empirically confirmed.

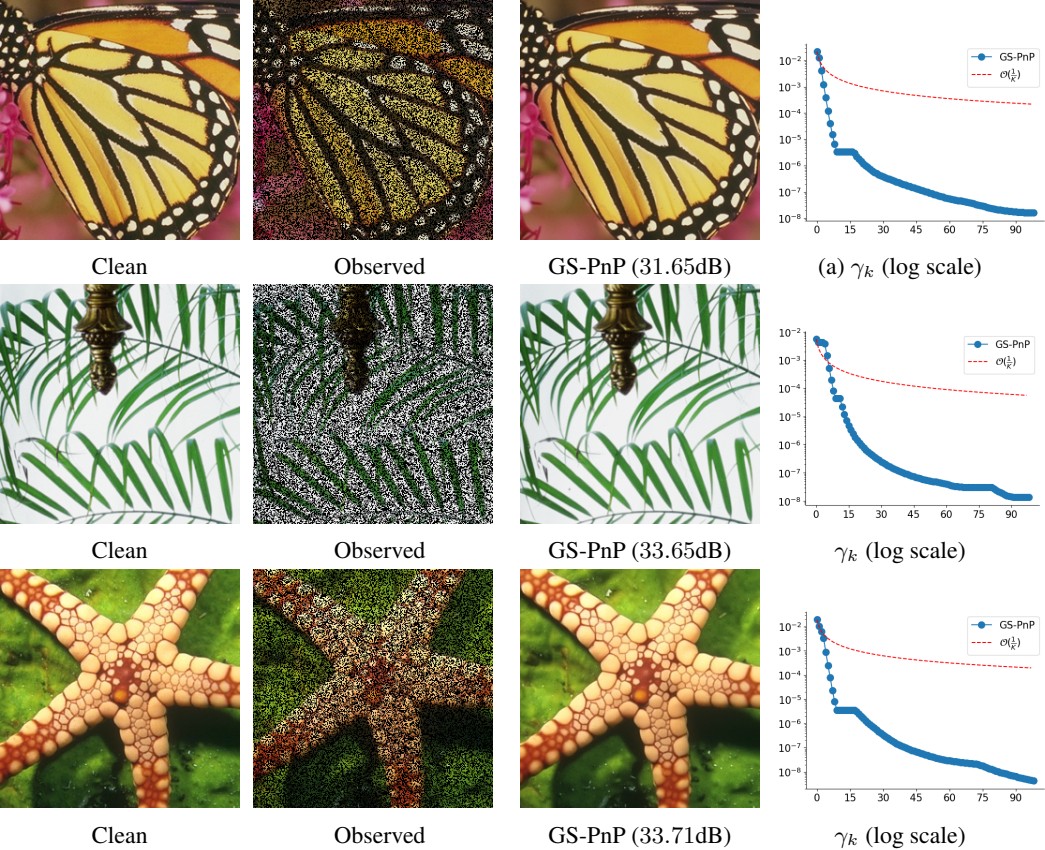

Figure 9: Inpainting results on set3C with pixels randomly masked with probability $p = 0.5$. In the last colomn, we show the evolution of $\gamma_k = \min_{0 \leq i \leq k} ||x_{i+1} - x_i||^2 / ||x_0||^2$ along the iterations.

## J.4 PSNR CONVERGENCE CURVES

We first plot Figure 10 the evolution of the PSNR along the iterations of the PnP algorithm during the experiments of Figure 1 and Figure 2. This illustrates that the minimization of $F$ coincides with the maximization of the PSNR, which supports the interest of the optimized functional $F = f + \lambda g_\sigma$.

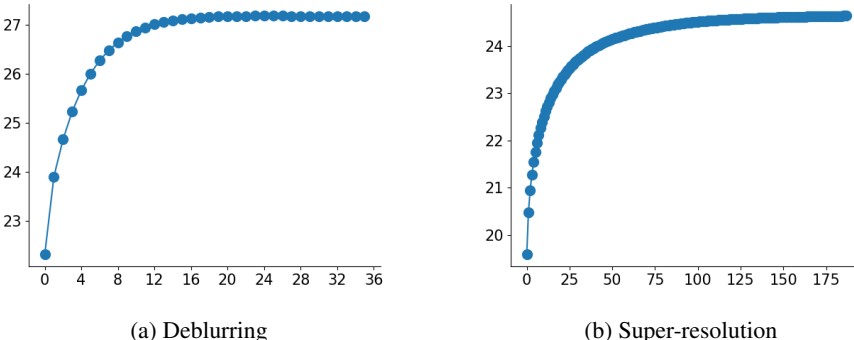

(a) Deblurring         (b) Super-resolution

Figure 10: Evolution of the PSNR along the iterations of the algorithm, during (a) the deblurring experiment of Figure 1 and (b) the super-resolution experiment of Figure 2. Note that the convergence in PSNR follows the convergence in function value (represented in Figures 1(g) and 2(g)).

## J.5 INFLUENCE OF THE PARAMETERS

In this section we study more deeply the influence of the parameters involved in the GS-PnP algorithm. Three parameters are involved: the stepsize $\tau$, the denoiser level $\sigma$ and the regularization parameter $\lambda$.

- The stepsize $\tau$ is automatically tuned with backtracking and is not tweaked heuristically, contrary to other competing methods based on PnP-HQS.
- The first regularization parameter $\sigma$ is linked to the used denoiser.
- The second regularization parameter $\lambda$ is introduced so as to target the objective function $f + \lambda g_\sigma$, which is the main purpose of our method. It is a classical formulation of inverse problems, and the trade-off parameter $\lambda$ is usually tuned manually.

Thus, like PnP-HQS, we have two parameters that we are free to tune manually. One additional motivation for keeping both $\lambda$ and $\sigma$ as regularization parameters is to be able to use our PnP algorithm with noise-blind denoisers like DnCNN that are independent on $\sigma$. In practice, in our experiments, we first roughly estimated $\sigma$ proportionally to the input noise level $\nu$ and tweaked $\lambda$ more precisely. Note that, for each inverse problem, our parameters $\lambda$ and $\sigma$ are fixed for a large variety of kernels, images and noise levels $\nu$. The parameters are not optimized for each image.

Figure 11 and Figure 12 respectively plot the average PSNR when deblurring the CBSD10 images with different values $\lambda_\nu$ and $\sigma/\nu$, and fixed $\nu = 0.03$. Both parameters control the strength of the regularization. We observe that $\lambda_\nu$ and $\sigma$ have a similar influence on the output: for small $\lambda_\nu$ or small $\sigma$, the regularization involved by the denoising pass is not sufficient to counteract the noise amplification done by the proximal steps with large $\tau$ (recall that when $\tau \to \infty$, $\mathrm{Prox}_{\tau f}$ tends to the pseudo-inverse of $A$). On the contrary, as expected, increasing $\lambda$ and $\sigma$ tends to over-smooth the output result.

On a single image, we also vary the main parameters of both GS-PnP and RED (Figure 13). For fair comparison, the PSNR is computed on the luminance channel only. This experiment confirms that, when manually optimizing the parameters for both methods, the PSNR results obtained with GS-PnP and RED remain close, as already observed in Table 6.

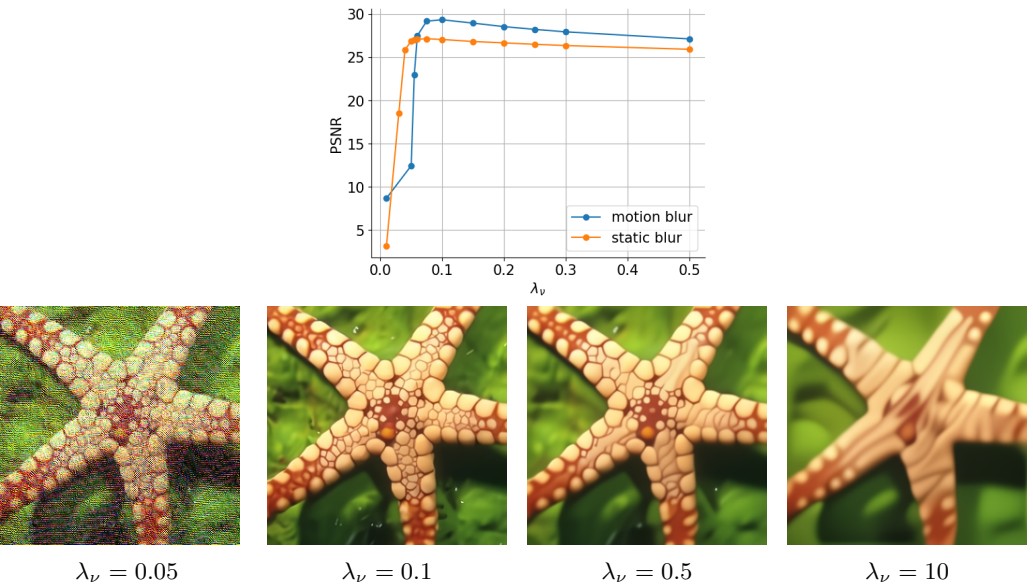

Figure 11: Influence of the choice of the parameter $\lambda_\nu$ for deblurring. Top: average PSNR when deblurring the images of CBSD10, blurred with motion blurs or static blurs, for different values of $\lambda_\nu$. The other parameters remain unchanged. Bottom: visual results when deblurring "starfish" with various $\lambda_\nu$ (in the same conditions as Figure 1).

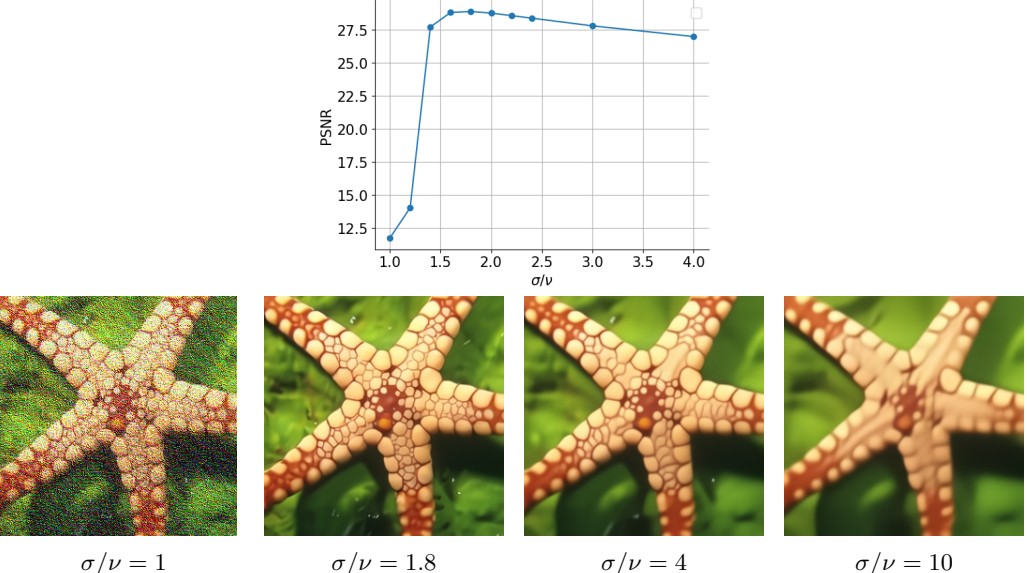

Figure 12: Influence of the choice of the parameter $\sigma$ for deblurring. Top: average PSNR when deblurring the images of CBSD10, blurred with the 10 kernels, for different values of $\sigma/\nu$, with $\nu = 0.03$. The other parameters remain unchanged. Bottom: visual results when deblurring "starfish" with various $\sigma/\nu$, with $\nu = 0.03$ (in the same conditions as Figure 1).

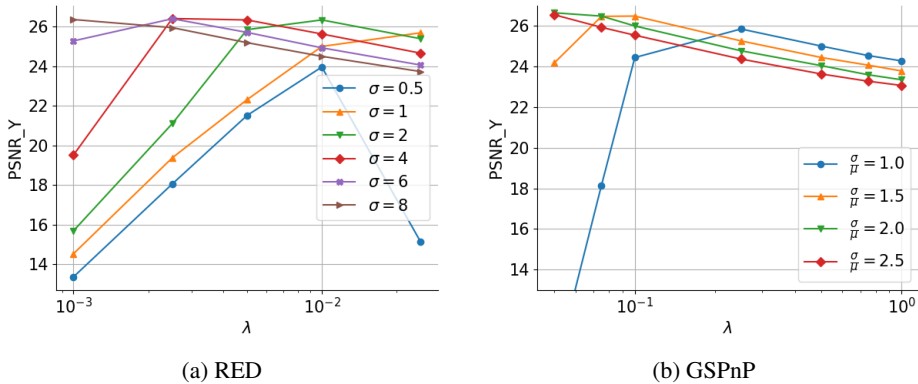

(a) RED  (b) GSPnP

Figure 13: Influence of the parameters $\sigma$ and $\lambda$ for RED and GS-PnP when deblurring the single image "starfish" degraded with uniform kernel and $\nu = 7.65/255$. For fair comparison, like in Table 6, the PSNR is calculated on the $Y$ channel only. Remember that the results of Table 6 were obtained with $\sigma = 3.25, \lambda = 0.02$ for RED and $\sigma = 2\nu, \lambda = 0.075$ for GS-PnP.

## J.6 INFLUENCE OF THE INITIALIZATION

In Figure 14, we examine the robustness of the method to the initialization. As can be seen on this experiment, the output image does not change much even for relatively large perturbation of the initialization. We thus observe a robustness to the initialization, both in terms of visual aspect and PSNR. We also observed that initializing with a uniform image does not change the output of the algorithm. We suggest that this robustness comes from the first proximal steps on the data-fidelity term (with a large $\tau$), which prevent the algorithm to be stuck in a poor local minimum. Note that the use of large $\tau$ in the beginning of the algorithm is possible thanks to the backtracking procedure.

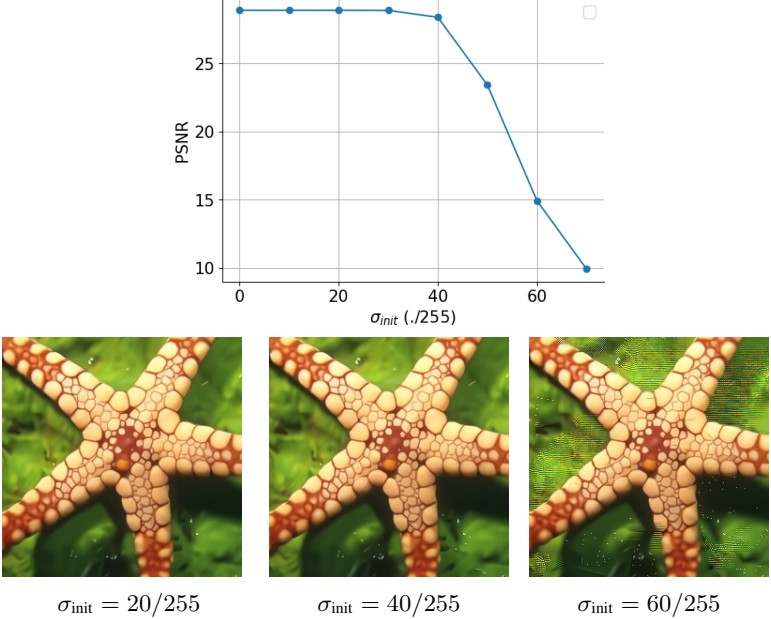

$\sigma_{\text{init}} = 20/255$  $\quad\quad$ $\sigma_{\text{init}} = 40/255$  $\quad\quad$ $\sigma_{\text{init}} = 60/255$

Figure 14: Influence of the initialitation $z_0$ on the deblurring result. Instead of initializing with the blurred image $z_0 = y$ as done in Section 5.2.1, we set $z_0 = y + \xi_{\sigma_{\text{init}}}$ with $\xi_{\sigma_{\text{init}}}$ an AWGN with standard deviation $\sigma_{\text{init}}$. By increasing the noise level $\sigma_{\text{init}}$, we investigate the robustness of the result to changes in the initialization of the algorithm. Top: PSNR values, along with values of $\sigma_{\text{init}}$. Bottom: corresponding visual results on "starfish" with various $\sigma_{\text{init}}$ (in the same conditions as Figure 1). The algorithm is robust to noisy initializations up to a relatively large value of $\sigma_{\text{init}}$.

## J.7 CONVERGENCE OF DPIR (ZHANG ET AL. (2021))

In this section, we illustrate that, contrary to our method, the DPIR algorithm is not guaranteed to converge and can even easily diverge. In Figure 15, we plot the convergence curves of both DPIR and our GS-PnP when deblurring the "starfish" image degraded with a motion kernel and $\nu = 0.01$. In the original DPIR paper Zhang et al. (2021), only 8 iterations are used with decreasing $\tau$ and $\sigma$. More precisely, $\sigma$ decreases uniformly in log-scale from 49 to the input noise level $\nu$, and $\tau$ is set proportional to $\sigma^2$. In order to study the asymptotic behaviour of the method, we propose two strategies to run DPIR with 1000 iterations:

(i) Decreasing $\sigma$ from 49 to $\nu$ over 1000 iterations instead of 8 (Figure 15, row 1).

(ii) Decreasing $\sigma$ from 49 to $\nu$ in 8 iterations, and then keep the last values of $\sigma$ and $\tau$ for the the remaining iterations (Figure 15, row 2).

As illustrated by the plot of $\sum_{i \leq k} ||x_{i+1} - x_i||^2$ (third column), DPIR fails to converge with both strategies, even if the residual $||x_{k+1} - x_k||^2$ tends to decrease with the second strategy. This divergence also involves a loss of restoration performance in terms of PSNR (first column). On the other hand, as theoretically shown in this paper, the residual $||x_{k+1} - x_k||^2$ with GS-PnP tends to 0 (reaches $\sim 10^{-13}$ before the activation of backtracking, versus $\sim 10^{-4}$ for DPIR) and its series converges.

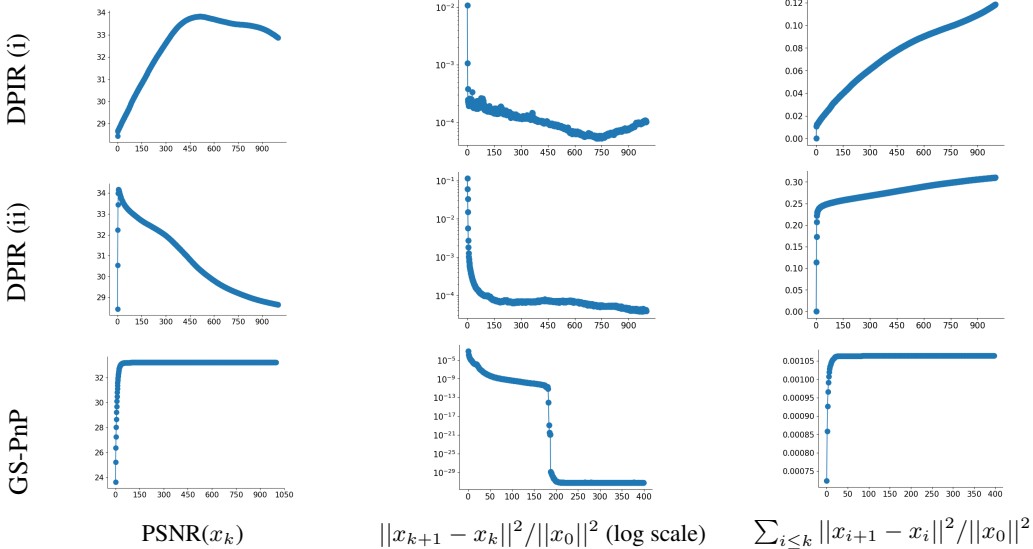

$$PSNR(x_k) \qquad ||x_{k+1} - x_k||^2/||x_0||^2 \text{ (log scale)} \qquad \sum_{i \leq k} ||x_{i+1} - x_i||^2/||x_0||^2$$

Figure 15: Convergence of the DPIR algorithm versus convergence of GS-PnP when deblurring the "starfish" image. The two first rows display results obtained with DPIR with two different strategies used for decreasing $\sigma$: in the first row, $\sigma$ is decreased from 49 to $\nu$ over 1000 iterations; in the second row, $\sigma$ is decreased in 8 iterations and then kept fixed for the remaining iterations.

