# OpenReview forum: "Gradient Step Denoiser for convergent Plug-and-Play"
_ICLR.cc/2022/Conference — ICLR 2022 Poster_

### Official Review · Reviewer_E8QG · 2021-11-03

**Correctness:** 3
**Technical Novelty And Significance:** 3
**Empirical Novelty And Significance:** 3
**Recommendation:** 6
**Confidence:** 4

**Main Review:**

**Weakness**

1. As stated, the proposed regularizer is similar to the alternate prior proposed in RED. What is the difference? Why the re-invention is new in the context of the paper.

2. The convergence analysis seems to follow the standard proof in the non-convex optimization literature when the explicit regularizer is given. To me, the analysis is a straightforward extension of the classic results. Could the authors elaborate more on the difficulties in the proof?

3. Could the authors explain more about how the network is trained? Since the output of the neural network is an image-size vector and the regularizer is $||x-N_\sigma(x)||_2^2$, it is a little weird to me if one forces $\nabla g$ to be the noise residual. Why not make $N_\sigma$ a denoiser and explain $\ell_2$-norm as accommodation of the Gaussian noise.

4. Various paper has proposed backtracking steps to determine the step-size ($\tau$ in GS-PnP). Perhaps a comparison and discussion are required.

5. The final performance is about the same as the state-of-the-art plug-and-play algorithm (DPIR). If the authors can explain the novelty of the proposed regularizer properly, this will no longer be a downside. However, if the work is not novel in terms of theory, a marginal/zero improvement in the empirical performance will be a weakness.

6. Given the linkage to RED, the current paper should provide a proper review of the progress of RED as well.

**Summary Of The Paper:**

This paper makes an extension of the plug-and-play framework by formulating an explicit regularizer $g(x)=||x - N_\sigma(x)||_2^2$ whose gradient $\nabla g$ corresponds to the noise residual $x-D_\sigma(x)$. By replacing the proximal of regularizer with this gradient step denoiser, the authors proposed the GS-PnP algorithm based on the half quadratic splitting algorithm. Since the explicit regularizer is known, a convergence analysis is hence established by assuming the Lipschitz continuity of $\nabla g$.

This paper is closely related to the recent trend of learning a regularizer functional by using deep neural networks. The difference between the existing literature and this paper is that the former trains a deep neural network to directly output a scalar value, while the latter trains a neural network to output an image-size vector and then envelopes it with a $\ell_2$-norm. In fact, the proposed regularizer shares the same formulation as the one stated in Sec 5.2 *"An Alternative Prior"* in Romano's RED paper. However, the difference between the two is not clearly stated in the paper.

**Strength**
1. An extension of the PnP with explicit regularizer formulated.
2. (Non-convex) convergence analysis under the common assumptions of the Lipschitz continuity of $\nabla g$
3. Extensive validation on image denoising, deblurring, and super-resolution.



**Summary Of The Review:**

In general, I vote for *weak rejection* based on the current evaluation of the paper.

—— After rebuttal ——
I now vote for acceptance.

---

> ### Author Response · Authors · 2021-11-11
> **Response to reviewer E8QG (1/2)**
>
> 1 -  The main difference between our regularizer and the one alternately proposed in RED is the following :
> - RED considers a generic given pretrained denoiser $D_\sigma : \mathbb{R}^n \to \mathbb{R}^n$, which is then associated with the regularizer $g_\sigma(x) = \frac{1}{2}||x-D_\sigma(x)||^2$ and used as such in IR problems.
> - In our method, we set $g_\sigma(x) = \frac{1}{2}||x-N_\sigma(x)||^2$ (with $N_\sigma : \mathbb{R}^n \to \mathbb{R}^n$ differentiable) and then we train the denoiser as  $D_\sigma = Id - \nabla g_\sigma$ with the loss function $||D_\sigma(x+\xi)-x||^2$ for clean images $x$ and AWGN $\xi$.
>
> With this new formulation, we are ensured that $D_\sigma = Id - \nabla g_\sigma$ is inherently a conservative vector field, without further assumptions on $N_\sigma$. Thanks to this relation, the (slightly modified) PnP-HQS becomes a proximal gradient descent (PGD). We can then make use of convergence results of the PGD algorithm in the nonconvex setting to show the convergence of PnP-HQS.
>
> In contrast to the original RED paper, we aimed at finding one setting of Plug-and-Play image restoration that allows for a convergence proof with sufficiently general hypotheses. For this purpose, we had to consider this very particular form of regularization, which was  mentioned in the 2017 RED paper but explicitly left aside. About convergence of RED and its variants, see also our answer on point 6 below.
>
> 2 - As  pointed out, the convergence proof follows standard arguments of the nonconvex optimization literature. The objective of this work is not to derive new theoretical convergence proofs for non-convex optimization algorithms but rather to apply existing convergence results in the context of PnP. However, in order to ensure a better understanding of the main ideas of the paper, we found it useful to present a direct proof of Theorem 1, adapted to our setting and with our notations. Also, it is necessary to derive the sufficient decrease condition of equation (22) in order to justify the backtracking strategy and to prove Proposition 1.
>
> 3 - Our denoiser $D_\sigma$ is trained by minimizing the loss function (7): $||D_\sigma(x+\xi)-x||^2$ for clean images $x$ and AWGN $\xi$. We propose to build $D_\sigma$ from a generic architecture $N_\sigma$, which can itself be an efficient denoiser. In this paper, the state-of-the-art denoising architecture DRUNet is used for $N_\sigma$, but other architectures could be considered. As defined in (6), our denoiser reads  $D_\sigma(x)=N_\sigma(x)+J_{N_\sigma(x)}^T(x-N_\sigma(x)).$
>
> It is important to notice that applying $D_\sigma$ exactly corresponds to realizing a gradient step $\mathsf{Id}-\nabla g_\sigma$ on the regularizer $g_\sigma(x)=||x-N_\sigma(x)||^2$. Therefore, we also get an explicit regularizer $g_\sigma$ as a byproduct of the denoiser learning. This is a novelty with respect to PnP literature where such $g_\sigma$ is not available.
>
>
> 4 - Thank you for the suggestion.
>
> Similar backtracking strategies to the one presented in Section 4.2 have been proposed in (Ochs et al., 2014, Beck, 2017, Chapter 10) for nonconvex PGD. Other backtracking procedures could be investigated in future work. For instance, (Li & Lin, 2015) ensures the same sufficient decrease property (11) with a line search initialized with the Barzilai-Borwein rule.
>
> We believe that the optimal tuning of the time step is out the scope of the current paper, whose contribution is to provide a convergent PnP method for general IR problems. We  nevertheless propose to include the above discussion in the appendix if necessary.

---

> ### Author Response · Authors · 2021-11-11
> **Response to reviewer E8QG (2/2)**
>
> 5 - Our method indeed presents similar qualitative results than DPIR, but it comes with a proof of convergence. DPIR does not have any convergence guarantee.
>
> More generally, in the existing literature, PnP approaches have one of the following limitations:
> - They are not able to provide proof of convergence when non strongly convex data-fidelity terms are involved (Ryu et al., 2019), which is the case of some classical IR problems such as deblurring, super-resolution or inpainting
> - They are restricted to (nearly) nonexpansive denoisers (Reehorst and Schniter, 2018; Ryu et al., 2019; Sun et al., 2019; Xu et al., 2020). But it has already been shown that imposing such hard Lipschitz constraints on a deep denoiser network alters its denoising performance (Bohra et al., 2021; Hertrich et al., 2021).
> - They show convergence of iterates thanks to decreasing time steps (Chan et al., 2016), but there is no characterization of the obtained solution (it is not a minima or a critical point of any functional)
>
> On the other hand, our method is proved to converge to a stationary point of an explicit functional including a non strongly convex data-fidelity term. It also relies on a (possibly expansive) denoiser that, although being contraint to be a conservative vector field, allows to produce state-of-the-art results for various ill-posed IR problems. We believe that this is a worthy contribution.
>
> 6 - Some follow-up works on RED (Schniter and Reehorst, 2018; Cohen et al., 2021) were mentioned in the "Related works" section. We here provide a more detailed discussion, that may be included in the appendix if necessary.
>
> The convergence of the original RED algorithm is discussed in (Schniter and Reehorst, 2018). The authors provide a convergence proof for RED-PG which requires the denoiser to be nonexpansive, which, again, is a restrictive hypothesis.
>
> The main idea of the RED-PRO method (Cohen et al., 2021) is to consider a constrained version of the image restoration problem: instead of including an explicit regularization in the functional, RED-PRO aims at minimizing the data-fidelity term on the set Fix(D) of fixed points of a generic denoiser D. The study is conducted under the *hypothesis that the denoiser is demicontractive, which implies that Fix(D) is convex, thus leading to a convex optimization problem*. They also show that the solution of this convex optimization problem can be approximated with a PnP-FBS scheme that alternates between two operators: a denoising step, and a gradient step on the data fidelity term. They provide convergence results under the hypothesis that these two operators have common fixed points or that their composition has a fixed point. However, let us mention that the completeness argument used in the corresponding proof remains unclear to us (see the proof of Theorem 4.4 in Appendix A.4 in Cohen et al., 2021). Also, the demicontractivity of a generic denoiser seems difficult to verify in practice. In the case of our denoiser $\mathsf{Id}-\nabla g_\sigma$, demicontractivity cannot be obtained through the co-coercivity of $\nabla g_{\sigma}$, which may not be satisfied for a non-convex $g_{\sigma}$. Finally, checking the existence of fixed points for the RED-PRO operator does not appear straightforward. In contrast, the fixed points of the GS-PnP operator are directly related to the stationary points of the global functional $F = f + \lambda g_\sigma$ (Lemma 1 in Appendix C), whose existence is guaranteed as soon as $F$ is coercive (see the discussion in Appendix D).

---

### Official Review · Reviewer_GZzY · 2021-11-03

**Correctness:** 3
**Technical Novelty And Significance:** 2
**Empirical Novelty And Significance:** 3
**Recommendation:** 6
**Confidence:** 4

**Main Review:**

I think the overall idea of this paper is great: designing a differentiable neural network, by construction, that equals the gradient descent step of an explicit regularizer function, the fact which provides new insights for solving an important open problem about the convergence of PnP and RED.  The idea sounds simple but promising and makes it possible to be directly applied for other variant PnP/RED algorithms. My main concerns / remark with this paper are as follows:

1). It is unclear for me why RED led to poor results for both deblurring and super-resolution. This may be due to the improper parameter tunning such as regularization trade-off, step-size and denoising intensity of the denoisers. Since the fixed points of PnP/RED algorithms are directly affected by the step size used in those algorithms [Ahmad et al., 2020, p.108], to show the true performance of DPIR and RED, the step size must be fine-tuned against PSNR. This can be done directly on the testing images. Additionally, the neural network architecture of denoisers used within RED need to be clarified.

2). To no loss of completeness, an additional interesting path to follow for this proposed method is to have a connection between convex regularization and CNNs, which would help to facilitate the optimization process and allows to derive recovery methods with global convergence. Hence, a brief discussion for providing insights of building convex $g_\sigma(x)$ through deep neural networks $N_\sigma$ is highly recommended.

3). Since the setups for $g_\sigma$ is not necessarily convex, resulting a non-convex problem. An initial point of the update sometimes is important. It would be interesting to see how different initialization influence the final reconstruction results.

4). In order to have a better understanding about how the regularization parameter $\lambda$ explicitly adjust the balance between the data-fit and the unconstrained deep neural network regularizer, it is useful to include a figure showing the evolution of images reconstructed by PnP-GS for different $\lambda$.

5). It is helpful to include the proposed GS-DRUNet and DRUNet light denoiser architecture in the supplementary materials to make this work easy to follow.

6). Having a convergent plot against PSNR (dB) would help to see the speed of improvement in imaging quality.

Minor comments:

1). Table (5) in the supplement, “31.70” is displayed in bold while “31.93”is not.

2). Some missing citations related to PnP:

[1] Buzzard et al., 2018, Plug-and-Play Unplugged: Optimization Free Reconstruction using Consensus Equilibrium.

[2] Ahmad et al., 2020, Plug-and-play methods for magnetic resonance imaging: Using denoisers for image recovery.


**Summary Of The Paper:**

The paper considers a novel and interesting idea: designing a deep neural network denoiser that makes Plug-and-play priors (PnP) convergence analysis clean and simple, motivated by PnP-HQS [Zhang et al., 2017b] and regularization by denoising (RED) [Romano et al., 2017]. Existing works have proved the convergence of PnP and RED with contractive and nonexpansiven denoisers. It is an activate area for designing deep neural net denoisers that ensure PnP/RED convergence. Specifically, this paper focuses on designing denoisers that correspond to the gradient descent step in terms of the $L$-smooth regularizer function $g_\sigma (x)$. By using such denoising step within PnP followed by the proximal of data fidelity $f$, one can guarantee convergence via traditional non-convex optimization given the objective function $F(x)$.  Since the handy $F(x)$ is available, a standard backtracking scheme is used in this work to ensure convergence without needs for tracking the exact Lipschitz constant of $\nabla g_\sigma$. Finally, the performance and stability of the proposed method is evaluated over three image inverse problems such as deblurring, super-resolution and inpainting, with satisfactory results compared to existing methods based on PnP and RED.

**Summary Of The Review:**

This paper is proposing an interesting alternative to training contractive/nonexpansiven denoiers for PnP and RED with convergence guarantee. Although I would expect a little bit more technical depth at least via numerical simulations, I believe this is a well-written paper with some new results that could spark further research in the important topic of designing explicit regularizers for PnP/RED.

---

> ### Author Response · Authors · 2021-11-11
> **Response to reviewer GZzY (1/2)**
>
> 1 - The main reason of the poor results obtained with RED is that the original RED algorithm only operates on a single channel (though we do not see any clear motivation for this, except to save computational time). Thus, it must be pointed out that in the deblurring and super-resolution experiments shown in both RED papers (Romano et al 2017, Cohen et al, 2021), the algorithm is applied only on the luminance channel, and PSNR values are computed likewise. For this reason, the chrominance channels are not denoised with RED and there exists a residual noise that explains the poor PSNR performance when computing errors on the 3 channels. For fair comparison, in Table 6 we reported PSNR results obtained with our GS-PnP algorithm applied on the luminance channel only, to exactly fit the methodology adopted in the RED papers. With this setting, the performance gap between RED and GS-PnP is reduced.
>
> About parameter tuning, let us mention that the proposed GS-PnP algorithm leads to competitive PSNR results *without* precise parameter tuning. In practical cases of image restoration (where the ground truth is unknown), manual tweaking of the parameters may be cumbersome. Thus it seems important to us to provide a set of parameters that can be used to treat successfully a large class of images. However, we agree that the sentivity of the results to the choice of parameters should be discussed. While it would require more time to do it on the whole database for all parameters and all methods, we added an experiment to highlight the impact of the parameters of GS-PnP on the PSNR result : $\lambda_\nu$ (Figure 11) and $\sigma$ (Figure 12).
>
> On a single image, we also varied the main parameters of both GS-PnP and RED (Figure 13). For fair comparison, the PSNR is computed on the luminance channel only. This experiment confirms that, when manually optimizing the parameters for both methods, the PSNR results obtained with GS-PnP and RED remain close, as already observed in Table 6.
>
> In the paper (Zhang et al, 2021) introducing DPIR, the authors have already discussed the tuning of the parameter $\lambda$, and exhibited the range [0.19, 0.55] which leads to favorable performance.
>
> The denoiser used within RED is the Trainable Nonlinear Reaction Diffusion (TNRD) denoiser, as proposed in the original paper. It has been specified in the paper.
>
> 2 - This is indeed an interesting suggestion that we discuss below.
>
> By building convex regularizers $g_\sigma$ with a deep neural network, the global convergence of the PnP schemes would follow from convex optimization results. Nevertheless, if we exclude simple linear denoisers $N_\sigma(x)=N_\sigma x$, it is a difficult challenge to define a class of network $N_\sigma$ such that $g_\sigma(x)=||x-N_\sigma(x)||^2$ is convex. Imposing convexity indeed requires to design a network $N_\sigma$ such that the Hessian matrix of $g_\sigma$ is non-negative for any $x$.
>
> A possibility to tackle this problem is to take inspiration from the literature of Convex Neural networks (Bach, 2021) or Input Convex Neural Networks (Amos et al., 2017) to learn a function $g_\sigma$ that is a convex function of (some of) its inputs. However, let us mention that our first attempts to directly model the regularizer $g_\sigma$ as a scalar neural network and to train the denoiser as $Id-\nabla g_\sigma$ (with the loss (8)) proved unsuccessful in terms of denoising performance (as mentioned at the beginning of Section 3.1). In our experiments, we only obtained relevant denoising results with the proposed formulation $D_\sigma(x)=N_\sigma(x)+J_{N_\sigma(x)}^T(x-N_\sigma(x))$, which allows to take advantage of existing denoising network architectures $N_\sigma$. For all these reasons, we suggest that it is unlikely that the denoising performance improves by imposing an aditional convexity constraint when  learning $g_\sigma$.
>
> This whole discussion could be included in the appendix.
>
> 3 - Indeed, for non-convex optimization algorithms, the sensitivity to the initialization should be discussed. In the supplementary material, we added a Figure 14 that examines the robustness to the initialization. As can be seen on this experiment, the output image does not change much even for relatively large perturbation of the initialization. We thus observe a robustness to the initialization, both in terms of visual aspect and PSNR. We also observed that initializing with a uniform image does not change the output of the algorithm.
> We suggest that the robustness to the initialization comes from the first proximal steps on the data-fidelity term (with a large $\tau$), which prevents the algorithm to be stuck in a poor local minimum. Note that it is the backtracking procedure that allows to use large $\tau$ in the beginning of the algorithm.

---

> ### Author Response · Authors · 2021-11-11
> **Response to reviewer GZzY (2/2)**
>
> 4 - We agree that the visual impact of the regularization parameter $\lambda$ is worthy of comment. We added in Figure 11 some deblurring results obtained with GS-PnP with various values of $\lambda$. As classically observed for inverse problems, the choice of parameter $\lambda$ may severly impact the success of the restoration method. However, in the paper we proposed a value of $\lambda$ (proportional to $\nu^2$) that seems to work well for a large class of images.
>
> 5 - The DRUNet architecture has been included in the supplementary material.
>
> 6 - Convergence plots against PSNR has been included in the appendix J.4, Figure 10.
>
> Also, your minor comments have been addressed and the references have been added to the discussion.

---

### Official Review · Reviewer_xYLt · 2021-11-05

**Correctness:** 4
**Technical Novelty And Significance:** 4
**Empirical Novelty And Significance:** 3
**Recommendation:** 8
**Confidence:** 4

**Main Review:**

## Pros.
I enjoy reading this paper -- the writing is so great, the theory proved is so strong and general, the empirical simulation is also extensive and convincing, making this paper one of the best papers I have ever reviewed.

The proposed gradient step denoiser formulation in the PnP framework is particularly interesting. By such a modification, they can reach a very general theoretical convergence result without relying on assumptions of strongly convex data terms $f$ and non-expansiveness on $g$ --- the best result obtained from prior research. Moreover, they claim the established theorem can even be extended to non-convex $f$ --- this is also super exciting.

## Cons.
I have some questions that need clarification:
- The main results are obtained based upon HQS algorithm. I'm wondering whether the theoretical results are still validated on other proximal algorithms, e.g., proximal gradient descent, ADMM, etc.
- I think in Remark 1, the gradient of $g_\sigma^*$ should be equivalent to the negative log gradient of the smoothed image prior $p_\sigma$, according to Eq.(3) in [R1].
- The gradient step formulation of denoiser requires the calculation of Jacobian (Eq. (6)), which makes the training (in Eq. (8)) a bit cumbersome. Could the authors provide more details on how they compute the Jacobian? How to do that with the PyTorch automatic
differentiation tools?
- Recent advances on PnP methods, e.g., [R2] should be mentioned in this paper.

[R1] Solving Linear Inverse Problems Using the Prior Implicit in a Denoiser, arxiv 2021
[R2] Tuning-free Plug-and-Play Proximal Algorithm for Inverse Imaging Problems, ICML 2020


**Summary Of The Paper:**

This paper presents a new form of the plug-and-play (PnP) half-quadratic splitting algorithm with provable convergence.
In contrast to directly formulating the denoiser as previous works, they propose to parameterize the denoiser with a learnable score-based function. This builds a bridge between the recently emerged score-based generative model and plug-and-play methods. And even more surprisingly, they show such a new formulation within the PnP framework leads to a very strong theoretical convergence guarantee under mild realistic assumptions. They also demonstrate good empirical results on three image restoration tasks i.e., deblurring, super-resolution and inpainting, verifying the empirical convergence rate is usually faster than $\mathcal{O}(\frac{1}{k})$ -- the worst-case rate established theoretically.

**Summary Of The Review:**

From my point of view, this is a truly innovative work with significance on both theoretical and practical sides. Since PnP is a very important tool in image reconstruction domains, this paper is thus worth being highlighted and would inspire further works in future.

Disclaimer: I don't check the full details of mathematical derivations. The justification here is therefore contingent upon the prior assumption: all the Theorem/Remarks in this paper are rigorously established.

---

> ### Author Response · Authors · 2021-11-11
> **Response to reviewer xYLt**
>
> > I enjoy reading this paper -- the writing is so great, the theory proved is so strong and general, the empirical simulation is also extensive and convincing, making this paper one of the best papers I have ever reviewed.
>
> Thank you for this positive feedback.
>
> > 1 -  The main results are obtained based upon HQS algorithm. I'm wondering whether the theoretical results are still validated on other proximal algorithms, e.g., proximal gradient descent, ADMM, etc.
>
> Unfortunately not. Our denoiser can be used in any PnP method (PnP-ADMM, PnP-DR...) but the proof of convergence we provide is currently only valid for the PnP-HQS  framework $prox_{\tau f} \circ D$, where $D$ is a denoiser. With this formulation, using our gradient step denoiser, PnP-HQS (with a relaxed denoising pass) corresponds to a proximal gradient descent (PGD or PnP-FBS) for the objective (10): $F=f+\lambda g_\sigma$. In the convex setting, the proof of convergence of PnP-ADMM use other properties (such as nonexpansiveness of reflexive proximal operators), which hinders a straightforward extension of our result.
>
> The convergence is obtained thanks to the PGD analysis in the nonconvex setting.
> In our case, given the regularity of our functions $f$ and $g_\sigma$, PGD seems to be the most adapted algorithm to optimize $f+\lambda g_\sigma$:
> - ADMM is useful when $f$ and $g_\sigma$ are not differentiable but proximable. With our formulation, $prox_{g_\sigma}$ is not tractable.
> - When the data fidelity term $f$ is not differentiable (see noise-free inpainting in Appendix J.3), Gradient Descent cannot be used.
>
> > 2 - I think in Remark 1, the gradient of $g_\sigma^*$ should be equivalent to the negative log gradient of the smoothed image prior , according to Eq.(3) in [R1].
>
> Yes, you are right. Thank you. This typo has been corrected (and we also corrected Tweedie's Identity at the top of Page 3 as $D_\sigma(x)=x+\sigma^2\nabla_xp_\sigma (x)$)
>
> > 3 - The gradient step formulation of denoiser requires the calculation of Jacobian (Eq. (6)), which makes the training (in Eq. (8)) a bit cumbersome. Could the authors provide more details on how they compute the Jacobian? How to do that with the PyTorch automatic differentiation tools?
>
> The Jacobian-vector product $J_{N_\sigma(x)}^T(x-N_\sigma(x))$ is automatically computed thanks to PyTorch automatic differentiation function `autograd.grad` :
>
> ```
> N = DRUNet_light(x,sigma)
> torch.autograd.grad(N, x, grad_outputs=x - N, create_graph=True, only_inputs=True)[0]
> ```
>
> Such computation is similar to what is done for training the discriminator $d:R^n\to R$ of a Wasserstein Generative model with gradient penalty (see the code on [github WGAN-GP](https://github.com/eriklindernoren/PyTorch-GAN/blob/master/implementations/wgan_gp/wgan_gp.py)), where the loss function includes a term $(||\nabla_x d(x)||_2-1)^2$.
>
> > 4) Recent advances on PnP methods, e.g., [R2] should be mentioned in this paper.
>
> We have added in the introduction of Section 5.2 a discussion concerning the tuning of the parameter $\lambda$, mentioning [R2]:
>
> *For each IR problem, we provide default values for the parameter $\lambda$ that can be used to treat successfully a large class of images. Performance can be marginally improved by tuning $\lambda$ for each image, for example with the method of (Wei et al., 2020) based on reinforcement learning.*

---

### Official Review · Reviewer_QQES · 2021-11-05

**Correctness:** 4
**Technical Novelty And Significance:** 2
**Empirical Novelty And Significance:** 2
**Recommendation:** 6
**Confidence:** 5

**Main Review:**


This work is well-written and easy to follow for the most parts.
It discusses a new denoiser to be employed within a plug-and-play framework, which indeed is a hot topic and an interest of the community. However, the current version of the work is not ready for publication due to following reasons.




**Primary issues:

1 – the proposed regularizer in (5) is obscure and no interpretation around itself is provided. What does it impose on the recovery?  Besides, $N_{\sigma}$ which seems to play the most important role in (5) is introduced only as “parametrized by a neural network”, and later it is restricted to be a differentiable neural net or “net with differentiable activation functions”; it is unclear what $N_{\sigma}$ does, and how it is related to (8) and Remark 1.



2 – The GS-PnP (9) is deviated from being a PnP-HQS while using (4) as the employed denoiser. The author(s) claimed it (“Departing from a slight modification of the PnP-HQS framework….”); however, the right-hand side of (9) seems like a heuristic approach which may lead to a better numerical result by adjusting parameters like $\lambda$ and $\tau$ which are not introduced in the denoiser (4).

The HQS for (1) is either $Prox_{\tau f} \circ Prox_{\lambda \tau g}$  or  $ Prox_{\lambda \tau g} \circ Prox_{\tau f} $, where PnP-HQS replaces  \Prox_{\lambda \tau g} with a denoiser $D$. The strength of the filter $D$ is then either determined by $\lambda$ and $\tau$ (and the sparsifying transform used within $D$) consistent with the analysis of proximal operator, or one can use a heuristic approach to set the filtering strength (or $\sigma$) through cross-validation. The results by the right-hand side of (9) might seem appealing (marginally) due to tweaking of the parameters $\lambda$, $\tau$ and $\sigma$.


3 – it is not clear whether the proposed GS denoiser (9) is limited to be employed within the HQS (half-quadratic splitting) method or can be used within other recovery frameworks such as ADMM and FBS. For instance, RED is independent of the recovery framework; is the GS (9) the same?





4 – the Proof of Theorem 1 (i): First of all it is not clear how (16) is derived, by which the rest are built upon. The (16) can be valid only when “$f(x)$” satisfies restricted strong convexity. While the authors claimed “this new PnP algorithm allows for non strongly convex data terms”.
“bounded and/or nonexpansive  denoiser”+ “decreasing step-size”+ “Lipschitz constraint”, “These settings are unrealistic as deep denoisers do not generally satisfy such properties.”
“This comes at the cost of imposing strong convexity on the data term f, which excludes many IR tasks like deblurring, super-resolution and inpainting”
“One strength of this approach is to simultaneously allow for a non strongly convex (and not smooth) data term …. .”
Nothing has been proved for the cases where “$f(x)$” is not strongly convex; however, the proof of theorem 1 is based on the strong convexity of “$f(x)$” and .



5– “we plug a denoiser that inherently satisfies equation 3 without sacrificing the denoising performance” + “But imposing hard Lipschitz constraints on the network alters its denoising performance.”
The main objection on these claims is that how the performance of the denoising is inspected?
Indeed, an stand alone denoising task is different from a general image restoration problem (e.g., deblurring, super-resolution, compressive sensing and inpainting) as the underlying degradation within each problem is different. Therefore, using the same denoiser for different image restoration problem is problematic, and expecting that a denoiser tuned for stand-alone AWGN denoising task works well for any IR task is unrealistic.


6 – it is not clear how the tuning is performed for the other methods used for the comparison. For instance in Table 2, how are the results obtained by other comparative methods (e.g., IRCNN and DPIR)? Did you use the same set of parameters tuned by their authors for CBSD64 over CBSD10?
“Due to large computational time of some compared methods ...” please ignore those slow methods (which I think is EPLL providing the least of results – or even RED in deblurring) and obtain the results for the whole CBSD64 for the sake of fair comparison and avoiding possible overfitting. This applies to other experiments too.

**Secondary issues:
1 – (2) is incorrect! It should be $Prox_{f}(z) = argmin_{x} \{ \frac{1}{2} ||x-z||_2^2 + f(x) \}$ …


2 – “Besides, this new PnP algorithm allows for non strongly convex data term, and thus can address ill-posed IR tasks ….” is indistinct and vague!! By “data term” do you mean “data-fidelity term”? If so, how and where non-convexity (or non-strongly convex!) data-fidelity is investigated?
“Besides, this new PnP allows….” Does it mean other PnPs do not allow such capability but the proposed one does?

“This comes at the cost of imposing strong convexity on the data term f, which excludes many IR tasks like ….” the same issue stated above!

3 – “Though providing excellent restorations, such schemes are not guaranteed to converge for all kinds of denoiser or IR tasks.” is this claim correct? How is it related to your work? Does it mean the proposed work is able to work with any denoiser and for any IR tasks (e.g., compressive sensing)?



4 – “contrary to Romano et al (2017), our denoiser exactly represents a conservative vector field.” any proof? Does it have a symmetric Jacobian?

5 – “inpainting noise-free input images lead to a non differentiable data term $f$”! why?

6 – what are the number of parameters in Table 1?


**Minor issues/modifications/suggestions:

1 – perhaps the regularization parameter in (1) should be positive, i.e. $\lambda>0$

2–  equation 4, equation (22), (6) … please use a consistent way of citing equations, e.g., use within parentheses.

3– The forward model in page can be formulated in a separate equation due to its frequent use throughout the paper.

4– HQS (half-quadratic splitting) is used but is not cited, please see https://doi.org/10.1109/83.392335

5–  “external denoiser” is vague! Perhaps “generic denoiser” is a better alternative.

6– “exact” proximal mapping is vague! Perhaps “explicit” proximal mapping.

7– “subsequent integration in PnP schemes” + “subsequent PnP image restoration” are vague/incomprehensible!

8– the first paragraph of page 3 is vague! It should be rewritten.

 “One can relate the deep denoiser” → “a deep”

 “The latter shows that the Minimum ….” what is the former?

 What are “$p$” and “$*$”? they are not introduced. Do they correspond respectively to density and convolution?

 “intractable” is vague!

9– “equation 4 comes down to including”

10– “Algorithm (9)” typo. Either (9) or Algorithm 1.

11– “is non-increasing and converging” → converging toward?

12– It would be better if only the average results are represented in Table 2, instead of so many numbers.

13– “Both IRCNN and DPIR realize PnP-HQS”

14– “i-e” typo. → “i.e.”

**Summary Of The Paper:**

The paper proposed a new denoiser, based on the gradient of a proposed trained regularizer, to be employed within a plug-and-play (PnP) framework, which is claimed to have a convergence proof compared to other existing methods (which claimed to use "unrealistic" assumptions). Experiments are provided to support the efficacy of the proposed method.

**Summary Of The Review:**

Post-rebuttal:

I thank the authors for their careful responses to the reviewers’ concerns, comments and questions. Their response addresses most of my concerns. I highly recommend to include the responses to the major issues in the paper to rectify the ambiguities, and re-write Section 3 accordingly. I would like to change my evaluation based on the authors' response, to accept.


-------------------------------------------------------------

In a nutshell, the submitted manuscript suffers from the following issues:
1) lacking a clear motivation/description on the proposed denoiser (4) and (5),
2) ambiguity of the proposed PnP method (9),
3) unsupported claims and lacking a clear proof for Theorem 1 by which the contribution of paper is built upon,
4) insufficient experiments,
which overall could not convinced me to accept the paper, but to evaluate it as “reject”.

---

> ### Author Response · Authors · 2021-11-11
> **Response to reviewer QQES (1/3)**
>
> ### Primary issues
>
> 1 - In order to propose a generic convergent PnP scheme, our main idea is to define the denoiser as a gradient descent operator: $D_\sigma(x)=x-\nabla g_\sigma(x)$ (equation (4)).
>
> A first interpretation for $g_\sigma$ is given in Remark 1, in relation with Tweedie's formula: after learning such a $D_\sigma$ with $L^2$ loss, one can interpret the energy $g_\sigma$ as an approximation of $-\sigma^2 \log p_\sigma$, where $p_\sigma$ is the smoothed true image prior.
>
> Different choices can be made for $g_\sigma$. As mentioned in the paper, we chose the formulation (5) that was already proven efficient in (Salimans  and Ho, 2021) for generative modeling with Energy Based Models. In this formulation, $N_\sigma:\mathbb{R}^n \to\mathbb{R}^n$ is a differentiable neural network, which takes as input an image and outputs an image of the same size.
>
> Our denoiser is finally defined in (6) by plugging (5) in (4) to obtain $D_\sigma(x)=N_\sigma(x)+J_{N_\sigma(x)}^T(x-N_\sigma(x)).$
>
> This leads to a second interpretation: the denoiser $D_\sigma$ is a corrected version of the neural network $N_\sigma$ with an additive term that makes $D_\sigma$ a conservative vector field.
> The neural network has to be differentiable to be able to evaluate the Jacobian operator $J_{N_\sigma}$ required in $D_\sigma$. Any differentiable neural network architecture can be used to parametrize  $N_\sigma$. In our experiments, we choose to use the DRUNet architecture.
>  We train the denoiser  $D_\sigma = Id - \nabla g_\sigma$ with the loss function $||D_\sigma(x+\xi)-x||^2$ for clean images $x$ and AWGN $\xi$. Relation (8) is this very same training loss expressed in terms of $g_\sigma$. It is obtained by plugging (4) in (7).
>
> 2 - As rightly pointed out, our PnP algorithm involves three parameters : the stepsize $\tau$, the denoiser level $\sigma$ and the regularization parameter $\lambda$.
> - The stepsize $\tau$ is automatically tuned with backtracking and is not tweaked heuristically, contrary to other competing methods based on PnP-HQS.
> - The first regularization parameter $\sigma$ is linked to the used denoiser.
> - The second regularization parameter $\lambda$ is introduced so as to target the objective function $f + \lambda g_\sigma$, which is the main purpose of our method. It is a classical formulation of inverse problems, and the trade-off parameter $\lambda$ is usually tuned manually.
>
> Thus, like PnP-HQS, we have only two parameters that we are free to tune manually.
>
> One additional motivation for keeping both $\lambda$ and $\sigma$ as regularization parameters is to be able to use our PnP algorithm with noise-blind denoisers like DnCNN that are independant on $\sigma$.
>
> In practice, in our experiments, we first roughly estimated $\sigma$ proportionally to the input noise level $\nu$ and tweaked $\lambda$ more precisely. Note that, for each inverse problems, our parameters $\lambda$ and $\sigma$ are fixed for a large variety of kernels, images, and noise levels $\nu$. They are not optimized for each experiment.
>
> 3 - Our denoiser could be used in any PnP method (PnP-ADMM, PnP-DR...), but our convergence proof is only valid for the PnP-HQS framework $prox_{\tau f} \circ D$, where $D$ is a denoiser. With this formulation, using our gradient step denoiser, PnP-HQS (with relaxed denoising pass) corresponds to a proximal gradient descent (PGD) for the objective (10) $F=f+\lambda g_\sigma$. The convergence is thus obtained thanks to the PGD analysis in the nonconvex setting.
>
> Note that, given the regularity of our functions $f$ and $g_\sigma$, PGD seems to be the most adapted algorithm to optimize $f+\lambda g_\sigma$:
> - ADMM is useful when $f$ and $g_\sigma$ are not differentiable but proximable. With our formulation, $prox_{g_\sigma}$ is not tractable.
> - When the data fidelity term $f$ is not differentiable (see noise-free inpainting in Appendix J.3), Gradient Descent cannot be used.
>
> 4 - There is no use of the strong convexity of $f$ in the proof of Theorem 1.
> Relation (16) is a definition that has no relation with the strong convexity of $f$ (note that (16) also involves $g_\sigma$). Relation (16) introduces the notation $Q_\tau(x)$ that includes $f(x)$, the linearization of the differentiable function $g_\sigma(x)$ around x and a coupling term $||x-y||^2$. Hence (16) is not related to the convexity property of $f$ or $g_\sigma$. The functional $Q_\tau$ is later used to show the convergence of the algorithm. This is a classical tool to show convergence of optimization algorithms in the nonconvex setting.

---

> ### Author Response · Authors · 2021-11-11
> **Response to reviewer QQES (2/3)**
>
> 5 - We would like to underline that using a denoiser tuned for AWGN denoising to solve various IR problems is a classical pipeline for PnP methods. This framework has already produced very good results in various IR tasks (see Yuan et al., 20Z0, or Zhang et al., 2021). Using the same denoiser for various IR tasks is precisely the main advantage of PnP methods as one does not need to retrain deep models for each individual task.
>
> Let us also mention that it was already empirically observed (Romano et al., 2017; Zhang et al, 2021) that the performance of the denoiser directly impacts the performance of the corresponding PnP scheme for IR.
>
> Also, similar to deep unfolding methods, we could learn to specialize our algorithm to each kind of degradation.  This is however out of the scope of the present paper.
>
> 6 - We indeed run the compared methods with the set of parameters given by the authors. Concerning the difference between CBSD68 and CBSD10, note that we provided in Table 5 the same comparison of deblurring performance (for IRCNN, DPIR and GS-PnP) on the whole CBSD68 dataset. As commented in the paper: " On CBSD10 (Table 2) or on CBSD68 (Table 5), we observed very similar performance gaps between the compared methods". We added in the new version of the paper, in Table 8, the same SR performance comparison (for IRCNN, DPIR and GS-PnP) on the whole CBSD68 dataset. The same observation can be done. This confirms that CBSD10 is large enough to compare fairly and accurately the PnP methods.
>
> ### Secondary issues
>
> 1 - Thank you for noticing the typo. It has been corrected.
>
> 2 - "data term" has been changed to "data-fidelity term". The quoted sentence was changed to emphasize that strong convexity is not required for the convergence of our PnP scheme:
>
> *This convergence guarantee does not require strong convexity of the data-fidelity term, thus encompassing ill-posed IR tasks like deblurring, super-resolution or inpainting.*
>
> Avoiding the need of strong convexity of the data-fidelity term is a main benefit of our approach (and it is crucial for many IR tasks, e.g. deblurring, super-resolution or inpainting). Indeed, in the PnP literature, the other convergence studies have one of the following limitations:
> - They are not able to provide proof of convergence when non strongly convex data-fidelity terms are involved (Ryu et al., 2019).
> - They are restricted to (nearly) nonexpansive denoisers (Reehorst and Schniter, 2018; Ryu et al., 2019; Sun et al., 2019; Xu et al., 2020). But it has already been shown that imposing such hard Lipschitz constraints on a deep denoiser network alters its denoising performance (Bohra et al., 2021; Hertrich et al., 2021).
> - They show convergence of iterates thanks to time steps decreasing to 0 (Chan et. al, 2016), but there is no characterization of the obtained solution (it is not a minima or a critical point of any functional)
>
> In contrast, our method is proved to converge to a stationary point of an explicit functional even for non strongly convex data-fidelity terms and with a possibly expansive denoiser. We did not investigate nonconvex data-fidelity terms in our experiments. However, our proof of convergence can encompass nonconvex data-fidelity terms, with the slight modifications indicated in Remark 2.
>
> 3 - Yes, this claim is correct for the same reasons. Other PnP approaches do have one of the three above limitations, whereas our method does not. Our method can indeed be applied for any IR task, in particular for compressed sensing. Results on compressed sensing will be given in future work. Let us precise that the proposed work is not able to work directly with "any" plugged denoiser. The denoiser has to be trained in the specific form of the gradient-descent step (4). However, as mentioned in the paper, any (differentiable) neural network previously used for denoising can parametrize $N_\sigma$ in (5).
>
> 4 - Our denoiser is defined in (6) as $D_\sigma(x)= x-\nabla g_\sigma(x).$
> It is a conservative vector field as it can be rewritten as $D_\sigma(x)=\nabla \left(\frac{1}{2}||x||^2 - g_\sigma (x) \right).$
>
> 5 - As we consider the whole image as the unknown, nonmasked pixels have fixed values for the inpainting problem. The data-fidelity term is thus the characteristic function of a convex set, i.e. $f(x)= 0$ if $Ax=y$ and $+\infty$ otherwise. Hence $f$ is a non differentiable function.
>
> 6 - Here are the number of trainable parameters for each model:
> FFDNET : 494 K / DNCNN : 668 K / DRUNET : 32.6 M  / DRUNET light : 17.0 M / GS-DRUNet : 17.0 M.
> They could be included in the appendix.

---

> ### Author Response · Authors · 2021-11-11
> **Response to reviewer QQES (3/3)**
>
> ### Minor issues:
>
> 1 - At this point, we can keep $\lambda \geq 0$ in order to include the unregularized case, which is worthy of interest.
>
> 2- The notation for referenced equations is now completely uniform throughout the document.
>
> 3 - Our proposed forward operator $T_{GS-PnP}$ is defined in relation (9) together with the scheme $x_{k+1}=T_{GS-PnP}(x_k)$ due to space limitation.
>
> 4 - The reference has been added.
>
> 5 - As suggested, we changed to "generic denoiser".
>
> 6 - As suggested, we changed to "explicit proximal mapping".
>
> 7 - The first sentence with "subsequent" was changed to :
>
> *State-of-the-art IR results are currently obtained with denoisers that are specifically designed to be integrated in PnP schemes*
>
> The second instance of "subsequent" has been removed.
>
> 8 -
> - We changed "the deep denoiser to the 'true' natural image prior" to "the ideal deep denoiser to the 'true' natural image prior $p$"
>
> - We have reformulated the sentence as "In (Efron, 2011), it is indeed shown that"
>
> - The notation $p$ refers to the true prior and is now introduced above.The smoothed prior $p_{\sigma}$ is now defined explicitly, which avoids the notation * for the convolution.
>
> - “intractable” has been changed to "cannot be computed explicitly"?
>
> 9 - "comes down to including" has been changed to "is equivalent to include"
>
> 10 - We changed "Algorithm (9)" to "scheme (9)"" or "iterative scheme (9)"
>
> 11 - We believe that it is not incorrect to say that a sequence converges without specifying the limit.
>
> 12 - We believe that providing all numerical results helps to understand how the methods behave with various blur kernels. Besides, these numbers could be later used by other authors for detailed comparison.
>
> 13 -  "Both IRCNN and DPIR realize PnP-HQS" has been changed to "Both IRCNN and DPIR use PnP-HQS"
>
> 14 -  Corrected.
>
>
> ### Summary Of The Review:
>
> > In a nutshell, the submitted manuscript suffers from the following issues:
> >
> > 1 - lacking a clear motivation/description on the proposed denoiser (4) and (5),
>
> Our motivation is to define a denoiser that acts as a gradient descent step. This simple yet new idea is the core of our contribution. It allows to rely on an explicit functional to minimize, and thus to show the convergence of PnP-HQS for various ill-posed IR problems.
>
> > 2 - ambiguity of the proposed PnP method (9),
>
> In the proposed PnP method (9), $\tau$ corresponds to an algorithmic time step (automatically tunned with backtracking) and $\lambda$ is a regularization parameter. This is a classical setting when dealing with optimization algorithms for IR problems.
>
> > 3 - unsupported claims and lacking a clear proof for Theorem 1 by which the contribution of paper is built upon,
>
> The proof of Theorem 1 is correct. It follows well-established works such as (Attouch et al., 2013).
> Relation (16) is a definition and it is not related to the strong convexity of $f$.
>
> > 4 - insufficient experiments, which overall could not convinced me to accept the paper, but to evaluate it as
> “reject”.
>
>
> Compared to the literature, we believe that our experiments are convicing, both quantitatively and qualitatively. We invite the reviewer to have a look at the reference PnP papers (Chan et al., 2016; Romano et al., 2017; Ryu et al., 2019; Zhang et al., 2021) and compare the results. For deblurring and super-resolution, we conducted experiments on a wider set of kernels and noise levels. In the new version of the paper, we also added new figures investigating the influence of the parameters and of the initialization.
> We point out that we propose a method with similar qualitative results than the reference method DPIR (Zhang et al., 2021), while providing a proof of convergence (recall that DPIR does not have any convergence guarantee). Hence we propose a state-of-the-art PnP method that is proved to converge for ill-posed IR problems. We believe that this is a worthy contribution.

---

### Author Response · Authors · 2021-11-11
**General comment on the rebuttal.**

We would like to thank all reviewers for their careful reading of our submission, and their helpful comments and suggestions. A new version of our paper has been uploaded in the submission system. This new version integrates most of your suggestions, corrected typos, and new tables and figures. These novelties are detailed in the answers to the reviewers.

---

### Decision · Program_Chairs · 2022-01-20

**Decision:**

Accept (Poster)

**Comment:**

The paper proposes a plug-and-play method for solving imaging problems. Plug-and-play methods use a denoiser to solve linear inverse problems. The paper proposes a plug-and-play method and uses convex optimization tools from analyzing proximal gradient methods to provide convergence guarantees. The algorithm is applied to a variety of inverse problems showing that the method works well.

After the discussion period, all four reviewers recommend acceptance.
- Reviewer QQES provided a detailed review and raised a few concerns including a clear motivation for and description of the denoiser, and unsupported claims, in particular related to a proof in the paper. The authors revised the paper and responded in length to the claims, in particular they detailed steps and assumptions related to the theorem in their response. As a response, the reviewer changed their score to accept.
- Reviewer xYLt strongly supports acceptance based on the strong theoretical results and a very good exposition.
- Reviewer GZzY likes the overall idea of the paper and raised a few minor concerns and questions, which were addressed by the authors.
- Finally, reviewer E8QG also appreciates the method, convergence analysis, and extensive validation. The reviewer also raised a few minor concerns and asked for clarification, and the response of the authors resolved those concerns.

Based on my own reading and based on the reviews, I recommend acceptance. The paper provides a variant of a plug-and-play method, proves interesting convergence results for the method, and has a strong experimental evaluation of the method. I encourage the authors to take the feedback of the reviewers into account, which they have done for the most part already, and it would also be interesting to see the performance of the method for compressive sensing problems.